# EMLoC: Emulator-based Memory-efficient Fine-tuning with LoRA Correction

**Hsi-Che Lin**[1]    **Yu-Chu Yu**[1]    **Kai-Po Chang**[1,2]    **Yu-Chiang Frank Wang**[1,2]

[1]Graduate Institute of Communication Engineering, National Taiwan University    [2]NVIDIA

r13942079@ntu.edu.tw    frankwang@nvidia.com

## Abstract

Open-source foundation models have seen rapid adoption and development, enabling powerful general-purpose capabilities across diverse domains. However, fine-tuning large foundation models for domain-specific or personalized tasks remains prohibitively expensive for most users due to the significant memory overhead beyond that of inference. We introduce EMLoC, an Emulator-based Memory-efficient fine-tuning framework with LoRA Correction, which enables model fine-tuning within the same memory budget required for inference. EMLoC constructs a task-specific light-weight emulator using activation-aware singular value decomposition (SVD) on a small downstream calibration set. Fine-tuning then is performed on this lightweight emulator via LoRA. To tackle the misalignment between the original model and the compressed emulator, we propose a novel compensation algorithm to correct the fine-tuned LoRA module, which thus can be merged into the original model for inference. EMLoC supports flexible compression ratios and standard training pipelines, making it adaptable to a wide range of applications. Extensive experiments demonstrate that EMLoC outperforms other baselines across multiple datasets and modalities. Moreover, without quantization, EMLoC enables fine-tuning of a 38B model, which originally required 95GB of memory, on a single 24GB consumer GPU—bringing efficient and practical model adaptation to individual users. Project Page: hsi-che-lin.github.io/EMLoC

## 1 Introduction

General-purpose foundation models have demonstrated impressive zero-shot capabilities across a wide range of benchmarks [2, 6, 8, 11, 38]. For real-world deployment, such as domain-specific tasks [22, 46] or personalized user behavior [28, 33] , further customization by fine-tuning is still required. However, fine-tuning typically incurs significantly more memory overhead than inference [30]. Consequently, if users have a fixed amount of available computing resources, they will be forced to choose between two unfavorable options. First, they can use a small model that fits within their memory budget for fine-tuning as shown in Fig. 1(a), but this sacrifices the emergent capabilities [41] of larger models and underutilizes hardware during inference. Alternatively, they can opt for a large model that fully utilizes resources during inference but exceeds memory limits for fine-tuning as shown in Fig. 1(b), making user-specific adaptation infeasible and potentially limiting performance in specialized applications. This paper addresses a central research question: Is it possible to design a fine-tuning strategy such that users can fine-tune a model under the same memory budget as inference?

The memory cost of fine-tuning can be broadly attributed to three components: optimizer states, intermediate activations, and the model parameters themselves, as marked in Fig. 1 with different colors. Initial efforts to reduce the memory usage of fine-tuning concentrated on the first two components. The first component, optimizer states, stores auxiliary information such as momentum and variance in the Adam optimizer [16] for each trainable parameter. This overhead can be

39th Conference on Neural Information Processing Systems (NeurIPS 2025).

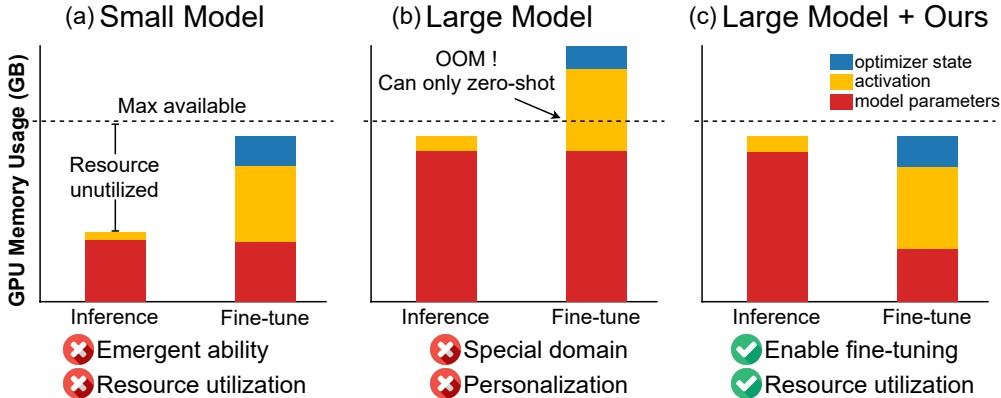

Figure 1: **The dilemma caused by additional memory overhead during fine-tuning. (a)** Users opt for a smaller 8B model, sacrificing emergent capabilities and underutilizing available hardware. **(b)** Use of a larger 26B model requiring memory exceeding the hardware limit even with LoRA [14] and gradient checkpointing [4] techniques. **(c)** Our EMLoC utilizes a smaller model during fine-tuning, allowing the same budget for both training and inference.

mitigated using parameter-efficient fine-tuning (PEFT) methods [14, 21, 45], such as LoRA [14], which reduces trainable parameters by only updating lightweight low-rank matrices. The second component, intermediate activations, refers to the values retained during the forward pass for use in backpropagation. Previous works reduced it by modifying model architectures [10, 20, 37] or gradient checkpointing [4]. However, since the components reduced by both strategies still presented during fine-tuning, reducing them alone, without addressing the memory usage of model parameters, cannot fully close the memory gap between fine-tuning and inference (Fig. 1(b)).

Few recent works have taken initial steps toward addressing the third component, the memory cost of model parameters. For example, Offsite-Tuning [43] proposes to drop some of the intermediate layers entirely during fine-tuning and to update only the top and bottom layers, thereby saving some memory occupied by model parameters. LORAM [49] introduces row-pruning as a method to reduce model parameters during fine-tuning, allowing the updating of the unpruned rows in the intermediate layers. Despite their promise, these methods typically rely on memory-intensive full-model training. This entails a strong and often impractical assumption that well-resourced institutions must be involved, making them unfeasible for individual users. In addition, they impose strict constraints on which subsets of weights can be updated, limiting their flexibility and general applicability across tasks and architectures. Moreover, they overlook the fact that different base models are used during fine-tuning and inference, introducing potential mismatches that degrade performance.

In this paper, we propose EMLoC, an Emulator-based Memory-efficient fine-tuning framework with LoRA Correction. EMLoC completely eliminates the memory cost gap between inference and training and can be carried out solely by individual users. The central idea is to perform fine-tuning on a low-rank model which we call the emulator so that the memory taken up by model parameters can be reduced as depicted in Fig. 1(c). To make this approach effective, we introduce an efficient, downstream-aware emulator construction procedure based on activation-aware singular value decomposition (SVD), using a small subset of downstream data for calibration. This yields a lightweight emulator tailored to the downstream data, without requiring costly full-model training. Fine-tuning is then conducted entirely on the emulator without any assumptions about the fine-tuning process—any standard training pipeline can be used. However, since the LoRA modules are fine-tuned on the emulator rather than the original model, naively merging the LoRA modules into the original model will suffer from the misalignment between the full model and the compressed emulator. To mitigate this issue, we further introduce a novel LoRA correction algorithm, which adjusts the parameters to compensate such discrepancy during inference.

Our contributions can be summarized as follows:

- We introduce a pipeline that closes the memory cost gap between inference and fine-tuning, enabling users to fine-tune a model under the same memory budge as inference.

- We propose a novel emulator-based fine-tuning framework that significantly reduces memory consumption by operating on a downstream-aware low-rank approximation of the model.

- We develop a LoRA correction algorithm that addresses the misalignment between emulator-based fine-tuning and full-model inference, improving final performance and transferability.

## 2 Related Works

### 2.1 Reducing Optimizer State

There are mainly two ways to reduce the memory required for optimizer states: by reducing the number of trainable parameters [12, 14, 15, 21, 45, 47, 50], or by reducing auxiliary information required per parameter [19, 34, 53]. A prominent example of the first approach is LoRA [14], which introduces trainable low-rank adapters into frozen pre-trained models to achieve parameter-efficient fine-tuning. For the second approach, Adafactor [34] eliminates the need to store full-rank second-moment estimators by using a factored approximation. While these methods are effective, a non-trivial number of trainable parameters are required to maintain acceptable performance. As a result, the optimizer states associated with those parameters continue to contribute a non-negligible portion of the overall memory footprint.

### 2.2 Reducing Intermediate Activation

Another way to fine-tune with less memory overhead is to reduce memory from intermediate activations. Two explored strategies are shortening the backpropagation path [27, 37, 45, 48] and recomputing activations during the backward pass [4, 10, 20]. An example of the first strategy is Ladder Side-Tuning [37], where trainable modules are placed outside the backbone to avoid back-propagation through the backbone. [20] carefully design the position and initialization of the adapter modules so that the pre-trained model becomes reversible, which enables recomputing intermediate activation from the output during backpropagation. However, these methods cannot completely eliminate activation memory due to the fundamental need for gradients, and they typically incur additional computational overhead.

### 2.3 Reducing Model Parameters

Beyond optimizing optimizer states and intermediate activations, very recently reducing the memory footprint of the model parameters during fine-tuning has been explored. It is worth noting that quantization-based methods [9, 52] are largely orthogonal to our focus: they do not reduce fine-tuning memory relative to inference, and can therefore be combined with our method. An approach to reducing parameter memory during fine-tuning involves using a pruned model for training while maintaining a full model for inference [43, 49]. For example, LORAM [49] assumes that the model publisher provides a row-pruned and continually pre-trained variant of the original model. Users fine-tune this reduced model and then transfer the learned LoRA modules back to the full model. However, their setting requires continual pre-training of the pruned model, making it inaccessible to individual users. Furthermore, only the unpruned weights can be fine-tuned, limiting flexibility in selecting which parameters to adapt. In contrast, our work introduces a pipeline that can be performed by users individually and allows flexible selection of tunable parameters.

## 3 Method

Our proposed method, EMLoC, is a memory-efficient fine-tuning framework that enables individual users to fine-tune a model with the same memory cost as inference. An overview of EMLoC is shown in Fig. 2. EMLoC consists of three main stages. First, we construct a downstream-aware lightweight emulator by performing activation-aware singular value decomposition (SVD) [40] on a small calibration set drawn from the downstream data. Second, we fine-tune the emulator with LoRA modules using any standard training pipeline. The reduced parameter size of the emulator is crucial for enabling memory-efficient fine-tuning. Finally, to address the misalignment issue between the original model and the emulator, we present a novel LoRA correction algorithm to correct the learned LoRA modules, and transfer them back to the original model before inference.

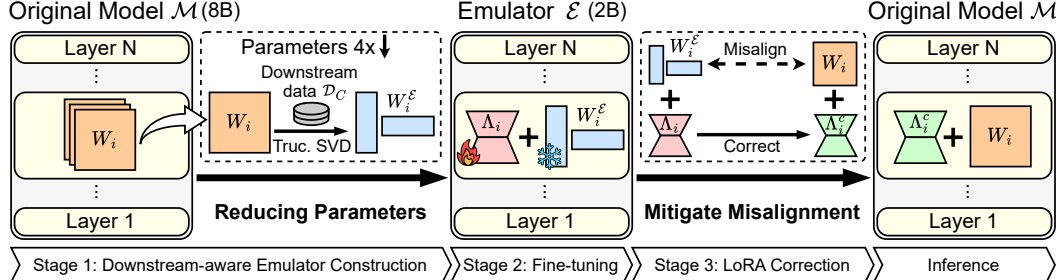

Figure 2: **Overview of EMLoC. Stage 1**: Construct a downstream-aware lightweight emulator. **Stage 2**: Fine-tune the emulator via LoRA, allowing reduced memory costs. **Stage 3**: Update the LoRA module to compensate the misalignment between the full model and emulator.

## 3.1 Memory-Efficient Fine-Tuning: Problem Formulation

Given a pre-trained model $\mathcal{M}$ and a downstream dataset $\mathcal{D}_{\text{down}}$, the goal is to fine-tune $\mathcal{M}$ with the same memory constraint as inference. We denote the collection of linear weights in $\mathcal{M}$ as $\mathbb{W} = \{W_i\}$. In EMLoC, we introduce an lightweight emulator, denoted as $\mathcal{E}$, constructed by replacing each $W_i \in \mathbb{W}$ with its counterpart $W_i^{\mathcal{E}} \in \mathbb{W}^{\mathcal{E}}$. The subscript $i$ here is retained to emphasize the positional correspondence between $W_i$ and $W_i^{\mathcal{E}}$ in the architecture. For simpleness, we may omit the index $i$ when the context is clear. Fine-tuning is then performed on the emulator $\mathcal{E}$ using any standard training pipeline. During training, a set of LoRA modules $\mathbf{\Lambda} = \{\Lambda_i\}$ is learned on top of $W^{\mathcal{E}}$, resulting in the fine-tuned model weights $W^{\mathcal{E}} + \Lambda$. At inference time, the learned LoRA modules are transferred back to the original model $\mathcal{M}$, with the corresponding weights $W + \Lambda$, which is used for inference. Note that conventional parameter-efficient fine-tuning (PEFT) methods are a special case of this framework, where $\mathbb{W}^{\mathcal{E}} = \mathbb{W}$ and hence $\mathcal{E} = \mathcal{M}$.

## 3.2 Downstream-aware Emulator Construction and Fine-tuning

**Downstream-aware Emulator Construction.** The goal of emulator construction is to create a lightweight replacement $\mathcal{E}$ for the original model $\mathcal{M}$ to be used during fine-tuning. To ensure that EMLoC is memory-efficient, generalizable, and effective, the emulator must satisfy three key criteria. First, it should have fewer parameters than the original model to reduce memory consumption during fine-tuning. Second, it should support flexible placement of LoRA modules, enabling all weights to be fine-tunable. In other words, if a LoRA module $\Lambda$ can be used to fine-tune a weight matrix $W$ in the original model, then EMLoC must support placing and training $\Lambda$ beside the corresponding $W^{\mathcal{E}}$ in the emulator, thereby enabling effective fine-tuning of $W$. Third, it should preserve knowledge relevant to the downstream task to ensure the fine-tuning process is effective. To meet all three criteria simultaneously, we propose constructing the emulator using activation-aware SVD, which naturally balances parameter reduction, flexibility, and task relevance.

Specifically, we construct the set of emulator weights $\mathbb{W}^{\mathcal{E}}$ by applying the activation-aware SVD method proposed in SVD-LLM [40] to each weight matrix $W \in \mathbb{W}$

$$\mathbb{W}^{\mathcal{E}} = \left\{ W^{\mathcal{E}} = W_U W_V = \text{SVD-LLM}(W, n) \mid W \in \mathbb{W} \right\}, \tag{1}$$

where $n$ is the number of kept singular values, chosen to control the parameter count of the emulator. The emulator $\mathcal{E}$ is then constructed by replacing each $W$ in $\mathcal{M}$ with its corresponding low-rank approximation $W^{\mathcal{E}}$. Although we denote each $W^{\mathcal{E}}$ as a single matrix, it is stored in its factored form $W_U W_V$, which significantly reduces the memory footprint of model parameters, thus satisfying the first criterion. Importantly, each $W^{\mathcal{E}}$ retains the same position, and the input/output dimensions as $W$, differing only in the rank of the transformation. As a result, any LoRA module $\Lambda$ that could be applied to $W$ can equivalently be applied to $W^{\mathcal{E}}$, enabling all weights to be fine-tunable, the second criterion. Finally, SVD-LLM minimizes the output reconstruction error

$$\|X^\top W - X^\top W^{\mathcal{E}}\|_F, \tag{2}$$

where $X$ is the intermediate activation computed from calibration data $\mathcal{D}_C$. This ensures that the emulator preserves task-relevant knowledge, making fine-tuning on it effective, thereby fulfilling the third criterion.

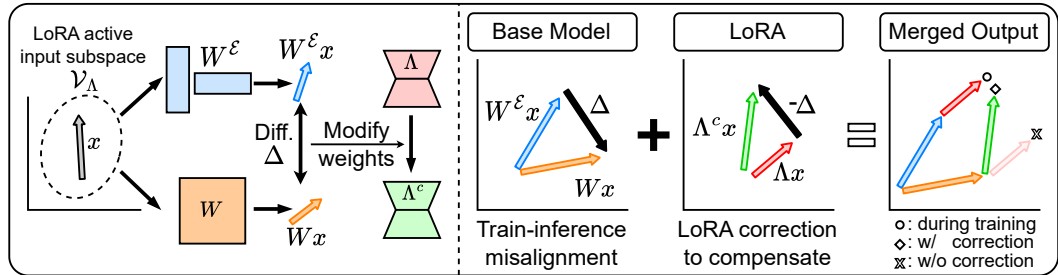

Figure 3: **LoRA correction to compensate model misalignment.** To alleviate the misalignment that arises from fine-tuning the lightweight emulator but running inference on the original model, LoRA parameters are corrected via feature spaces between the emulator and the original model.

**Fine-tuning.** Owing to the specific design choices in the emulator, we can apply any standard PEFT techniques [14, 45, 47] to fine-tune the significantly smaller emulator, achieving memory-efficient fine-tuning. In particular, if we aim to fine-tune a weight matrix $W_i$ in the original model, we instead insert a LoRA module next to $W_i^{\mathcal{E}}$ to fine-tune the emulator as in conventional frameworks. This allows us to use any existing training pipeline without modification, making our method a plug-and-play replacement for conventional LoRA fine-tuning, while further reducing memory usage.

### 3.3 LoRA Correction for Compression Misalignment

To fully utilize the capacity of the original model during inference, we aim to transfer the LoRA modules $\Lambda$ fine-tuned on $\mathcal{E}$ back to the original model $\mathcal{M}$. However, the emulator construction step that replaces $W$ with its low-rank counterpart $W^{\mathcal{E}}$ inevitably introduces a misalignment between these two models. As as result, the base model outputs seen by the LoRA modules are different during fine-tuning and inference. The difference is unexpected by the LoRA modules since they were optimized in the context of $W^{\mathcal{E}}$. To address this issue, we propose a novel LoRA correction algorithm and correct the misalignment explicitly, as shown in Fig. 3.

---

**Algorithm 1** LoRA_correction

**Input:** $W_A, W_B$: LoRA weights;
$W, W^{\mathcal{E}}$: Linear weights from $\mathcal{M}$ and $\mathcal{E}$;
$\lambda$: An hyperparameter for regularization.
**Output:** $W_A^c, W_B^c$: Corrected LoRA.
1. Apply SVD to $W_A$, obtaining $U, \Sigma, V^T$
2. $W_A' \leftarrow U, W_B' \leftarrow \Sigma V^T W_B$
3. $\Delta \leftarrow W_A'^{\top}(W - W^{\mathcal{E}})$
4. $W_B^c \leftarrow W_A'$
5. $W_B^c \leftarrow W_B' - \text{clamp}(\Delta, \lambda)$
**return** $W_A^c, W_B^c$

---

**Objective of LoRA correction.** The goal of our LoRA correction procedure is to correct the LoRA module $\Lambda$ fine-tuned with the emulator $\mathcal{E}$, and derive a new LoRA module $\Lambda^c$ suitable for the original model $\mathcal{M}$. In other word, for each input $x$ that activates the LoRA, the merged output should remain consistent between training and inference. Formally, we enforce the following condition:

$$x^{\top}(W + \Lambda^c) = x^{\top}(W^{\mathcal{E}} + \Lambda) \quad \forall x \in \mathcal{V}_{\Lambda}, \tag{3}$$

where $\mathcal{V}_{\Lambda}$ denotes the subspace on which $\Lambda$ produces none-zero outputs—i.e., the region where it is active. It is worth noting that we enforce Eq. (3) only within $\mathcal{V}_{\Lambda}$, so that the correction has effects exclusively when the LoRA module is active. This design prevents unintended side effects in parts of the model unrelated to the LoRA module when doing correction, thereby preserving the integrity of the original model's behavior.

Rewriting Eq. (3), we obtain an equivalent condition,

$$x^{\top}(\Lambda^c - \Lambda) = x^{\top}(W - W^{\mathcal{E}}) = \Delta_x \quad \forall x \in \mathcal{V}_{\Lambda}. \tag{4}$$

This formulation highlights the explicit correction objective. The desired $\Lambda^c$ aims to compensate for the misalignment $\Delta_x$, which exists between the outputs of $W$ and $W^{\mathcal{E}}$. This can be achieved through offsetting the original LoRA module by the correction term $\Delta_x$.

**Mitigating compression misalignment.** Although this idea is conceptually straightforward, two practical challenges arise. First, the condition in Eq. (4) must hold for all inputs $x \in \mathcal{V}_\Lambda$. Second, there is no direct way to incorporate the correction term $\Delta_x$ into the LoRA weights $\Lambda = W_A W_B$.

The first challenge is resolved by leveraging two facts: the condition in Eq. (4) is linear, and $\mathcal{V}_\Lambda$ forms a subspace of the input feature space. As a result, it suffices to enforce the condition only on a basis of $\mathcal{V}_\Lambda$, denoted by $\beta_\Lambda$

$$x^\top (\Lambda^c - \Lambda) = x^\top (W - W^\mathcal{E}) = \Delta_x \quad \forall x \in \beta_\Lambda. \tag{5}$$

Furthermore, a natural choice for $\beta_\Lambda$ is the set of column vectors of $W_A$, the input projection matrix of the original LoRA module. To address the second challenge, we introduce a preprocessing step that reparameterizes the LoRA module while preserving its functionality. Specifically, we perform SVD on the input projection $W_A$, yielding $W_A = U\Sigma V^\top$. We then redefine the LoRA factors as

$$W_A' = U, \quad W_B' = \Sigma V^\top W_B, \tag{6}$$

such that the overall transformation remains unchanged $\Lambda = W_A W_B = W_A' W_B'$. After this reparameterization, the columns of $W_A'$, denoted by $\{a_1, \ldots, a_r\}$, are orthonormal. This orthogonality makes the behavior of the LoRA module on each basis vector $a_i$ governed directly by the corresponding $b_i$, $i$-th row of $W_B'$.

$$a_i^\top \Lambda = \left( a_i^\top W_A' \right) W_B' = e_i^\top W_B' = b_i, \tag{7}$$

where $e_i$ is the standard basis vector. In other words, modifying the output of $\Lambda$ on each basis vector $a_i$ reduces to adjusting the corresponding row $b_i$ in $W_B'$.

With the challenges addressed, we now formally describe the LoRA correction algorithm, as presented in Algorithm 1. We begin by preprocessing the original LoRA module $\Lambda = W_A W_B$ to obtain $W_A'$ and $W_B'$. Next, we select the columns of $W_A'$ as the basis $\beta_\Lambda$ in Eq. (3). The correction term $\Delta$ is then computed by measuring the output difference between the original weight $W$ and the emulator weight $W^\mathcal{E}$ under this basis

$$\Delta = W_A'^\top \left( W - W^\mathcal{E} \right). \tag{8}$$

Each row of $\Delta$ represents the correction $\Delta_{a_i}$ corresponding to a basis vector $a_i$. Finally, we adjust $W_B'$ to satisfy the condition in Eq. (5), compensating for the misalignment between $W$ and $W^\mathcal{E}$

$$W_A^c = W_A', \quad W_B^c = W_B' - \text{clamp}(\Delta, \lambda), \tag{9}$$

where the clamp function limits the norm of each correction vector $\Delta_{a_i}$ to be no more than a multiple $\lambda$ of the corresponding $b_i$. This hyperparameter $\lambda$ acts as a safeguard against excessive deviation, and we find it improves robustness in practice. After applying Algorithm 1 to each LoRA module independently, the resulting corrected weights, $\Lambda^c = W_A^c W_B^c$, is thus adapted to account for the misalignment introduced by training on the emulator and inferring on the original model.

## 4 Experiment

### 4.1 Experiment Setup

**Datasets.** In our main experiments, we evaluate EMLoC on seven visual question answering (VQA) datasets. Four of these are widely used benchmarks: ChartQA [24], DocVQA [25], InfoVQA [26], and TextVQA [36]. To reflect realistic deployment scenarios that require domain adaptation, we also include three datasets from specialized domains: PMC-VQA [51] (medical imaging), WebSRC [5] (web page understanding), and WC-VQA [42] (multicultural knowledge). Ablation studies and additional analyses are primarily conducted on these three domain-specific datasets. For language modeling, we evaluate on two challenging mathematical reasoning benchmarks: MathQA [1] and GSM8K [7]. All reported metrics across datasets are based on accuracy.

**Comparison.** For the main experiments, we compare EMLoC against four baselines. **Original**: We directly fine-tune the original large model, InternVL2.5-8B [6], with LoRA. This baseline incurs significantly higher memory cost compared to other methods and serves as an upper bound in performance. **Small**: We fine-tune smaller variants of InternVL (4B and 2B) with LoRA, representing the low-resource alternative illustrated in Fig. 1(a). **Offsite**: We follow the Offsite-tuning [43] framework using the variant that omits full-model distillation to ensure fair comparison under the same compute constraints. **UPop**: A row-pruning-based method [35] is used to construct the emulator, followed by LoRA transfer using the procedure described in LORAM [49]. Except for the **Small** baseline, all methods—including ours—perform inference using the original InternVL2.5-8B model.

Table 1: **Performance comparisons of finetuning approaches on VQA with different memory costs.** With InternVL2.5-8B as the original model, and 50% indicates finetuning is conducted on 4B models. Note that direct finetuning on InternVL2.5-8B (i.e., rows in gray) exceeds the memory budgets of all other finetuning methods and thus is not feasible under the same constraints.

| Ratio | Method | ChartQA | DocVQA | InfoVQA | TextVQA | PMC-VQA | WebSRC | WC-VQA |
|-------|--------|---------|--------|---------|---------|---------|--------|--------|
| 50% | Original | 84.5 | 92.2 | 69.8 | 79.6 | 52.9 | 87.4 | 53.4 |
| | InternVL 4B | 82.9 | 90.4 | 64.6 | 75.8 | 50.6 | 84.8 | 48.6 |
| | Offsite [43] | 84.3 | 91.3 | 69.0 | 77.7 | 51.0 | 76.1 | 45.9 |
| | UPop [35] | 84.4 | 92.0 | 69.7 | 78.5 | 50.7 | 76.4 | 42.1 |
| | EMLoC | **84.6** | **92.3** | **69.9** | **78.8** | **52.3** | **85.2** | **48.8** |
| 25% | Original | 84.5 | 92.2 | 69.8 | 79.6 | 52.9 | 87.4 | 53.4 |
| | InternVL 2B | 78.8 | 87.4 | 56.0 | 73.2 | 44.6 | 78.1 | 34.4 |
| | Offsite [43] | 84.0 | **92.3** | 69.6 | 77.2 | 50.6 | 76.6 | 45.4 |
| | UPop [35] | **84.3** | 92.2 | 69.6 | 78.2 | 50.9 | 76.6 | 44.1 |
| | EMLoC | 84.2 | **92.3** | **70.0** | **79.0** | **51.6** | **79.6** | **46.2** |
| 25%+ QLoRA | Original | 84.8 | 92.0 | 68.0 | 79.5 | 52.3 | 84.8 | 51.9 |
| | InternVL 2B | 78.2 | 85.7 | 53.5 | 71.7 | 44.2 | 77.4 | 33.0 |
| | Offsite [43] | 83.9 | 91.8 | 67.0 | 77.1 | 50.1 | 75.2 | 44.2 |
| | UPop [35] | **84.2** | **91.9** | 67.1 | 78.1 | 50.3 | 75.8 | 43.9 |
| | EMLoC | **84.2** | **91.9** | **67.5** | **79.0** | **50.9** | **78.1** | **44.8** |

**Implementation details.** For EMLoC, we use LoRA with rank 8 and train with 500 iterations. The learning rate is set to $4 \times 10^{-5}$ with cosine annealing. The number of calibration data is set to 64 and $\lambda$ for LoRA correction is 3. The other baselines use the same hyperparameters setting except we adjust the LoRA rank so that the number of trainable parameters of all baselines are the same.

## 4.2 Main Results

**Main Results.** We evaluate EMLoC using InternVL2.5-8B as the base model under three realistic memory-constrained fine-tuning scenarios. These settings simulate user environments with training memory budgets equivalent to: fine-tuning a 4B model (denoted as 50% compression ratio in the table), fine-tuning a 2B model (25%), and fine-tuning a 2B model using QLoRA [9] quantization (25%+QLoRA). As shown in Table 1, EMLoC consistently outperforms all baselines across both standard and domain-specific VQA benchmarks. The improvements are especially significant on specialized datasets, where adaptation to the target domain is essential. Notably, EMLoC achieves performance closest to the upper bound set by full-model fine-tuning.

These results underscore the adaptability of our approach under varying memory constraints, maintaining strong performance even under the tightest budgets. In contrast to the common strategy of fine-tuning smaller models, EMLoC provides a more effective alternative by better leveraging the capacity of the full model. In addition, comparisons with UPop and Offsite-tuning demonstrate the advantages of our downstream-aware emulator construction. Unlike row-pruning or layer-dropping approaches that restrict architectural flexibility, our method tailors the emulator to downstream data without altering model structure, enabling more effective and generalizable adaptation. Finally, we note that EMLoC is orthogonal with existing techniques, such as gradient checkpointing [4] and quantization [9]. In our experiments, EMLoC is used together with these techniques, yet it consistently outperforms other baselines. This demonstrates that EMLoC is compatible with existing memory reduction methods, and can be integrated with them to achieve further efficiency gains.

## 4.3 Analysis

**Scalability on Model Size.** We further evaluate EMLoC on larger-scale models, including the 26B and 38B variants of InternVL2.5. For this experiment, we construct a 5B-sized emulator, making the fine-tuning cost roughly equivalent to training a 5B model. We use LoRA rank 6 for the 26B model and rank 4 for the 38B model. As shown in Table 2, EMLoC consistently improves upon the zero-shot performance of these large models, demonstrating its effectiveness even at greater model scales. Importantly, all fine-tuning were conducted with less than 24GB of GPU memory—substantially

Table 2: **Applying EMLoC to 26B and 38B models.** Note that a 5B-sized emulator is considered, and all experiments are conducted under a 24GB memory budget.

| Size | Method | PMC-VQA | WebSRC | WC-VQA |
|------|--------|---------|--------|--------|
| 26B | Zero-shot | 49.9 | 77.1 | 51.0 |
|     | EMLoC | **52.5** | **80.9** | **52.6** |
| 38B | Zero-shot | 52.5 | 79.0 | 53.6 |
|     | EMLoC | **57.0** | **82.1** | **56.8** |

Table 3: **Applying EMLoC to NLP tasks.** All settings follow LORAM [49], and a LLaMA2 13B is used as the full model.

|            | MATHQA | GSM8K |
|------------|--------|-------|
| w/o FT     | 32.6   | 24.3  |
| LORAM-RAND | 33.8   | 27.2  |
| LORAM-STRU | 33.8   | 24.6  |
| EMLoC      | **33.9** | **29.8** |

Table 4: **Comparisons of emulator construction settings.** The first row corresponding to LO-RAM [49] setting and the last row corresponding to EMLoC setting. Note that, LORAM requires external data and additional continual pretraining for memory efficient finetuning. We validate that both row pruning and SVD can be applied for our setting with better performances on various tasks using only a small subset of downstream data.

| Compression Method | Add. Data for compression | Require Add. Cont. Pre-train | Overhead ($\downarrow$) (GPU-hours) | PMC-VQA | WebSRC | WC-VQA |
|--------------------|---------------------------|------------------------------|-------------------------------------|---------|--------|--------|
| Row-pruning | Yes | Yes | 214 | 51.0 | **78.7** | 43.6 |
|             | No  | No  | **1.7** | **51.1** | 76.6 | **44.1** |
| SVD | Yes | Yes | 230 | 51.3 | 74.6 | 42.9 |
|     | No  | No  | **0.3** | **51.6** | **79.6** | **46.2** |

lower than the 95GB typically required for directly fine-tuning the 38B model. This means that users capable of running inference can also perform fine-tuning using our method, without additional hardware requirements. These results underscore the practicality and scalability of EMLoC for resource-constrained users.

**Comparison of Emulator Construction Settings.** We compare two emulator construction settings. The first is our proposed downstream-aware construction, in which the emulator is derived directly from task-specific data without any additional training. The second follows the approach adopted in LORAM, where the emulator is first constructed using general-purpose data, followed by continual pretraining. For general data, we use Laion-COCO [18]—a commonly used dataset for vision-language pretraining—and perform continual pretraining using 8 V100 GPUs.

As shown in Table 4, when row pruning is used as the emulator construction method, both settings achieve comparable downstream performance. However, the continual pretraining stage required in LORAM incurs substantial computational overhead, making it impractical for individual users. In contrast, when SVD is used for emulator construction, our downstream-aware method not only eliminates the need for continual pretraining but also achieves superior performance while incurring the lowest construction overhead, demonstrating the efficiency and effectiveness of our downstream-aware SVD. Furthermore, the performance gap observed in the last two rows highlights the importance of using downstream data rather than a general purpose data as the calibration set $\mathcal{D}_C$. By employing activation-aware SVD based on downstream activations, the constructed emulator better preserves task-relevant knowledge that facilitates subsequent fine-tuning.

**Robustness Across Modalities.** To evaluate the generality of our approach beyond vision-language tasks, we apply EMLoC to two challenging NLP datasets, MATHQA and GSM8K. We follow the experimental setup of LORAM and use LLaMA 2 13B [39] as the base model. As shown in Table 3, our method achieves comparable performance on MATHQA and outperforms LORAM on GSM8K, demonstrating its effectiveness in language-only settings. These results confirm that EMLoC is not limited to multimodal models and can be readily extended to other modalities. Furthermore, our method retains its core advantages—minimal memory overhead and no need for continual pretraining—making it more accessible to users with limited computational resources.

**Effectiveness of Transferring LoRA From Emulator to Original Model.** To better understand the role of the emulator and the effectiveness of our LoRA correction strategy, we compare model performance before and after transferring the fine-tuned LoRA modules from the emulator to the

Table 5: **Performance comparisons using finetuned emulators and EMLoC.** Note with while both finetuned emulator and EMLoC share the same memory budget, the former does not fully utilize such budget during inference and thus is not desirable.

| Ratio | Method | ChartQA | DocVQA | InfoVQA | TextVQA | PMC-VQA | WebSRC | WC-VQA |
|-------|--------|---------|--------|---------|---------|---------|--------|--------|
| 50% | Emulator | 57.1 | 62.3 | 29.3 | 15.0 | 40.9 | 42.1 | 30.7 |
| | EMLoC | **84.6** | **92.3** | **69.9** | **78.8** | **52.3** | **85.2** | **48.8** |
| 25% | Emulator | 19.3 | 6.3 | 10.8 | 6.9 | 31.2 | 4.0 | 19.1 |
| | EMLoC | **84.2** | **92.3** | **70.0** | **79.0** | **51.6** | **79.6** | **46.2** |
| 25%+ | Emulator | 14.8 | 4.5 | 10.5 | 7.3 | 32.2 | 4.3 | 19.0 |
| QLoRA | EMLoC | **84.2** | **91.9** | **67.5** | **79.0** | **50.9** | **78.1** | **44.8** |

Table 6: **Ablation study on activation-aware SVD and LoRA correction.**

| Ratio | Activation-Aware SVD | LoRA Correction | PMC-VQA | WebSRC | WC-VQA |
|-------|----------------------|-----------------|---------|--------|--------|
| 25% | ✗ | ✗ | 51.0 | 74.4 | 44.7 |
| | ✗ | ✓ | 51.2 | 74.4 | 44.8 |
| | ✓ | ✗ | 51.5 | 79.0 | 45.8 |
| | ✓ | ✓ | **51.6** | **79.6** | **46.2** |
| 50% | ✓ | ✗ | 51.8 | 83.7 | 47.9 |
| | ✓ | ✓ | 52.3 | 84.9 | 48.8 |

original model. As shown in Table 5, performance on the emulator itself is significantly lower the final model after LoRA transfer. This gap underscores the importance of using the full model for inference and justifies our design choice of transferring LoRA back.

Interestingly, despite the poor performance of the emulator, the LoRA modules trained on it improve the original model's downstream performance once transferred. This indicates that the fine-tuning process successfully extracts domain-specific knowledge, which is embedded in the LoRA weights and effectively transferred to the original model. These results highlight the effectiveness of our emulator-based training.

**Ablation Study.** We conduct ablation experiments to evaluate the contribution of two key components in the EMLoC framework: activation-aware SVD for downstream-aware emulator construction, and the LoRA correction algorithm for mitigating training-inference misalignment. As shown in Table 6, removing either component results in a drop in performance. This highlights the importance of preserving downstream-relevant knowledge during emulator construction, and the necessity of correcting for discrepancies introduced by training and inference on different models.

By further examining these results, we observe that the effectiveness of LoRA correction seems to be closely related to the degree of misalignment between the emulator and the original model. When the emulator is constructed without activation-aware SVD, it fails to retain key downstream behaviors of the original model, leading to larger misalignment during fine-tuning and consequently weaker correction effects. Conversely, under milder compression settings where misalignment is reduced, the impact of LoRA correction becomes more pronounced. These findings suggest that while our current use of SVD-LLM [40] provides a strong foundation for emulator construction, there remains room to explore more adaptive or task-aware compression strategies that could further enhance model alignment and correction effectiveness.

**Applying to Diffusion Model Personalization** To further assess the generalizability of EMLoC beyond text generation, we apply it to the task of personalizing a large text-to-image diffusion model. Specifically, we use DreamBooth [31]—a well-established personalization method—combined with LoRA to fine-tune FLUX.1-dev [17], a 12B state-of-the-art diffusion model. We conduct experiments on 8 subjects from the DreamBooth dataset, fine-tuning each for 500 steps without prior preservation. For evaluation, we report standard metrics used in prior works: DINO [3], CLIP-I [29], and CLIP-T scores. Due to the short fine-tuning duration in this experiment, we adopt standard SVD for emulator construction instead of activation-aware SVD, which would be less cost-effective in this setting.

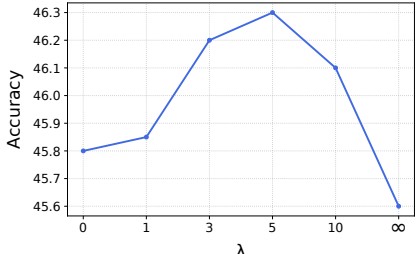

Figure 4: **Sensitivity analysis of $\lambda$ in the LoRA correction algorithm.** We plot performance on WC-VQA under different $\lambda$.

Table 7: **Quantitative results of applying EMLoC to diffusion model.** EMLoC achieves comparable, sometimes slightly lower, performance than direct fine-tuning while significantly reducing memory usage.

| Method | Fine-tuning memory (GB) | DINO | CLIP-I | CLIP-T |
|---|---|---|---|---|
| w/o EMLoC | 35.1 | **0.652** | **0.851** | 0.306 |
| w/ EMLoC | 22.9 | 0.615 | 0.831 | **0.321** |

Quantitative results are presented in Table 7. Note that normal fine-tuning requires over 35GB of memory, whereas EMLoC enables fine-tuning within a 24GB GPU budget. While EMLoC achieves slightly lower DINO and CLIP-I scores, it yields a higher CLIP-T score—likely due to slower convergence. We also show qualitative examples in Fig. 6. These results demonstrate that EMLoC can be effectively applied to image generation tasks, reinforcing its potential as a plug-and-play replacement for conventional LoRA fine-tuning with reduced memory requirements.

**Impact of Hyperparameter $\lambda$.** We investigate the effect of the hyperparameter $\lambda$, which constrains the norm of the correction term $\Delta$ in our LoRA correction algorithm. As shown in Fig. 4, performance varies with different choices of $\lambda$. In particular, when no constraint is applied (i.e., $\lambda \to \infty$), performance degrades noticeably. We observe that, in such cases, the correction term may have a significantly larger norm than the original LoRA output. We hypothesize that correction term with large norm will dominate and distort the LoRA modules, ultimately harming downstream performance. In contrast, applying an appropriate constraint on the correction magnitude helps preserve the learned knowledge in the LoRA modules while still compensating for model misalignment.

## 5    Conclusion

We present EMLoC, a novel framework that makes fine-tuning large foundation models as memory-efficient as inference. By introducing a downstream-aware emulator constructed via activation-aware SVD and performing all training on this low-rank approximation, EMLoC enables individual users to fine-tune large models without exceeding the memory budget required for inference. Our LoRA correction algorithm further improve the performance by addressing the mismatch introduced by training and inference on different model bases. Extensive experiments across diverse datasets and modalities demonstrate that EMLoC not only outperforms other baselines but also scales to models as large as 38B parameters on commodity hardware. These results open up new opportunities for accessible and scalable model customization, empowering broader adoption of large language models in specialized, resource-constrained settings.

**Acknowledgment**    This work is supported in part by the National Science and Technology Council via grant NSTC 113-2634-F-002-005, NSTC 114-2221-E-002-056-MY2 and NSTC 114-2640-E-002-006, and the financial supports from the Featured Area Research Center Program within the framework of the Higher Education Sprout Project by the Ministry of Education (114L900902). We also thank the National Center for High-performance Computing (NCHC) for providing computational and storage resources.

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

# EMLoC: Emulator-based Memory-efficient Fine-tuning with LoRA Correction

## Appendix

This supplementary document provides additional context, analysis, and results to support the main paper. We begin with a discussion of the limitations of our approach and potential directions for future work, followed by a brief examination of the broader social impact and additional related works. We then present supplementary proofs and derivations to justify components of our method. Additional implementation details are included to facilitate reproducibility. Finally, we provide extended experimental results that further validate the effectiveness and versatility of EMLoC.

## 6 Limitation and Future Work

While our emulator construction method enables significant memory savings and offers flexibility in placing LoRA modules, it relies on off-the-shelf SVD methods. These methods are primarily designed to preserve the inference-time behavior of the model, such as maintaining output logits, rather than the fine-tuning dynamics. As a result, the emulator may not fully capture the training characteristics of the original model, which could introduce suboptimalities during fine-tuning. This limitation is shared by all baselines. A promising direction for future work is to develop emulator construction strategies explicitly tailored to preserve fine-tuning behavior, such as gradient directions or parameter update trajectories. Such an approach could more faithfully mimic the training dynamics of the original model, potentially narrowing the performance gap between EMLoC and directing LoRA fine-tuning the original model.

## 7 Social Impact

EMLoC lowers the memory barrier for fine-tuning large foundation models, broadening access for users with limited resources and promoting more inclusive participation in AI development. While this democratization has clear benefits, it may also increase the risk of misuse, such as facilitating the generation of harmful or biased content. Additionally, although EMLoC reduces memory usage, widespread fine-tuning of large models may still contribute to environmental concerns. We encourage responsible use and transparency in deployment to mitigate these risks.

## 8 Additional Related Work

**Module arithmetic.** At a high level, EMLoC shares some conceptual similarities with module arithmetic [23, 32]. Both approaches involve modifying or adjusting model weights. While there is a conceptual similarity, our LoRA correction method serves a different purpose. Module arithmetic techniques typically aim to merge multiple modules to combine different capabilities. In contrast, our method focuses on addressing the discrepancy between the compressed emulator used during training and the full model used during inference, by compensating for the misalignment introduced by this difference to further improve the performance.

**SVD-related structure.** EMLoC may look similar with other SVD-related structure design [44]. LoRand can be viewed as a variant of the Houlsby adapter [13] that uses an SVD-like formulation to parameterize the up- and down-projection matrices. Specifically, it represents adapter weights as a product of summation of two low-rank matrices and a kernel matrix (i.e., $W = \sum P^T K Q$). By

adjusting the rank and sharing kernel matrices across layers, LoRand further reduces the number of trainable parameters. Similar to the distinction between EMLoC and LoRA, LoRand focuses on reducing the size of trainable adapter modules, whereas EMLoC reduces the memory for the pretrained weights in the base model, which improves the efficiency in a way not considered by LoRand. These approaches are orthogonal and complementary.

# 9 Supplementary Proofs and Derivations

## 9.1 Verification of Corrected LoRA

In the main text, we proposed that the corrected LoRA module $\Lambda^c = W_A^c W_B^c$ can be constructed by

$$W_A^c = W_A', \quad W_B^c = W_B' - \Delta, \tag{10}$$

where $\Delta = W_A'^\top (W - W^{\mathcal{E}})$. We now verify that substituting $\Lambda^c$ into the original model indeed satisfies the desired condition

$$x^\top (W + \Lambda^c) = x^\top (W^{\mathcal{E}} + \Lambda) \quad \forall x \in \mathcal{V}_\Lambda. \tag{11}$$

Note that we can write any $x \in \mathcal{V}_\Lambda$ as a linear combination of the columns of $W_A'$

$$\exists \gamma \in \mathbb{R}^d \text{ such that } x = W_A' \gamma \quad \forall x \in \mathcal{V}_\Lambda. \tag{12}$$

We now substitute this into the left-hand side of Eq. (11) and simplify:

$$\begin{aligned}
x^\top (W + \Lambda^c) &= (W_A' \gamma)^\top (W + \Lambda^c) \\
&= (W_A' \gamma)^\top (W + W_A^c W_B^c) \\
&= (W_A' \gamma)^\top (W + W_A' W_B' - W_A' \Delta) \\
&= (W_A' \gamma)^\top (W + \Lambda - W_A' W_A'^\top (W - W^{\mathcal{E}})) \\
&= \gamma^\top (W_A'^\top W + W_A'^\top \Lambda - (W_A'^\top W_A') W_A'^\top (W - W^{\mathcal{E}})) \\
&= \gamma^\top (W_A'^\top W + W_A'^\top \Lambda - I W_A'^\top (W - W^{\mathcal{E}})) \\
&= \gamma^\top (W_A'^\top W^{\mathcal{E}} + W_A'^\top \Lambda) \\
&= (W_A' \gamma)^\top (W^{\mathcal{E}} + \Lambda) \\
&= x^\top (W^{\mathcal{E}} + \Lambda) \quad \forall x \in \mathcal{V}_\Lambda,
\end{aligned} \tag{13}$$

which is exactly the right-hand side of Eq. (11).

## 9.2 Justification for Using SVD in LoRA Correction Preprocessing

In our LoRA correction algorithm, we introduce a preprocessing step where we apply SVD to the LoRA weights to obtain $W_A'$ and $W_B'$. It is important to note that, in the subsequent derivation of the correction procedure, the only essential property of $W_A'$ is that its columns form an orthonormal basis. Any other pair $(W_A'', W_B'')$ that satisfies this orthonormality condition can also be used in the correction algorithm. In this section, we show that although SVD may appear to be an arbitrary choice, it is a valid and convenient one — and critically, the final corrected LoRA result remains the same for any orthonormal decomposition.

Formally, suppose we have

$$W_A' W_B' = W_A'' W_B'' = W_A W_B, \tag{14}$$

where both $W_A'$ and $W_A''$ have orthonormal columns. Our goal is to show that the resulting corrected LoRA modules are identical:

$$W_A' \left( W_B' - W_A'^\top (W - W^{\mathcal{E}}) \right) = W_A'' \left( W_B'' - W_A''^\top (W - W^{\mathcal{E}}) \right). \tag{15}$$

Given Eq. (14), this reduces to prove that

$$W_A' W_A'^\top = W_A'' W_A''^\top. \tag{16}$$

Since the columns of both $W_A'$ and $W_A''$ are orthonormal and span the same subspace $\mathcal{V}_\Lambda$, $W_A' W_A'^\top$ and $W_A'' W_A''^\top$ are identical—they both represent the projection onto $\mathcal{V}_\Lambda$. Therefore, the resulting corrected LoRA modules are equivalent, regardless of the specific orthonormal basis chosen. This justifies the use of SVD in our preprocessing step: while not strictly necessary, it provides one convenient and consistent way to obtain the required orthonormal basis.

Table 8: Information of experiments for main results.

| Ratio | Method | Base Model Parameters | LoRA Rank | Trainable Parameters |
|---|---|---|---|---|
| 100% | Original | 8.1B | 8 | 18.9M |
| 50% | InternVL 4B | 3.7B | 10 | 18.7M |
| | Offsite | 3.8B | 64 | 18.9M |
| | UPop | 3.8B | 14 | 18.7M |
| | EMLoC | 3.7B | 8 | 18.9M |
| 25% | InternVL 2B | 2.2B | 19 | 18.7M |
| | Offsite | 2.2B | 64 | 18.9M |
| | UPop | 2.3B | 19 | 18.9M |
| | EMLoC | 2.3B | 8 | 18.9M |

Table 9: Information of the experiments for 26B and 38B large models.

| | InternVL-26B | InternVL2.5-38B |
|---|---|---|
| Original Parmaeters | 25.5B | 38.4B |
| Emulator Parameters | 5.0B | 5.3B |
| LoRA Rank | 6 | 4 |
| Trainable Parameters | 27.1M | 33.5M |

Table 10: **Performance comparison between EMLoC and zero-shot inference.** Note that fine-tuning with EMLoC requires less memory than performing inference. This enables users who can run inference to also benefit from fine-tuning, gaining additional performance improvements without increased memory cost.

| Quantization | Method | ChartQA | DocVQA | InfoVQA | TextVQA | PMC-VQA | WebSRC | WC-VQA |
|---|---|---|---|---|---|---|---|---|
| ✗ | Zero-shot | 83.8 | 92.0 | 69.6 | 78.5 | 48.6 | 76.5 | 43.1 |
| | EMLoC 50% | **84.6** | **92.3** | 69.9 | 78.8 | **52.3** | **85.2** | **48.8** |
| | EMLoC 25% | 84.2 | **92.3** | **70.0** | **79.0** | 51.6 | 79.6 | 46.2 |
| ✓ | Zero-shot | 83.7 | 91.3 | 67.1 | 78.6 | 48.1 | 67.4 | 42.0 |
| | EMLoC 25% | **84.2** | **91.9** | **67.5** | **79.0** | **50.9** | **78.1** | **44.8** |

# 10 Additional Implementation Details

## 10.1 Usage of Existing Assets

In our experiments, we make use of publicly available datasets, models, and codebases in accordance with their respective licenses. Specifically, we use the InfoVQA and TextVQA datasets, which are released under the CC-BY license; the PMC-VQA and WC-VQA datasets, available under the CC BY-SA license; and the WebSRC and LAION-COCO datasets, released under the MIT license; and the ChartQA data set, release under the GPL-3.0 license. The model backbones and training code are based on InternVL2.5, which is licensed under MIT. All assets were used strictly for academic, non-commercial research in accordance with their intended terms. We thank the respective authors and communities for making these valuable resources publicly available.

## 10.2 Implementation Details

**Main Results.** Detailed information, including the number of base model parameters, LoRA ranks, and the number of trainable parameters, for the experiments in our main results (Table 1) can be found in Table 8. Note that we use different LoRA ranks for each baseline to ensure that the number of trainable parameters is approximately the same across methods.

**Large Models.** Details of the experiments involving the 26B and 38B models (Table 2) are provided in Table 9. Note that for these experiments, we report performance on a subset of the test set due to limited computational resources, which made full-set inference infeasible within a reasonable time.

# 11 Additional Results

## 11.1 Zero-shot Results

In Table 10, we report the zero-shot performance of the backbone models used in our experiments. As expected, fine-tuning with EMLoC leads to consistent improvements over the zero-shot baselines.

Table 11: **Computational resource usage and performance comparisons of finetuning approaches.**
Ratios follow the definitions in Table 1. Note that EMLoC achieves stronger performance than other
baselines while maintaining comparable computation resource.

| Ratio | Method | Fine-tuning memory (GB) | Inference memory (GB) | TFLOPs | Throughput (sample/sec) | PMC | WebSRC | WC-VQA |
|-------|--------|------------------------|----------------------|--------|------------------------|------|--------|--------|
| 100% | Original | 22.3 | 16.1 | 37.4 | 0.09 | 52.9 | 87.4 | 53.4 |
| 50% | InternVL 4B | 15.4 | 8.1 | 18.9 | 0.17 | 50.6 | 84.8 | 48.6 |
| | Offsite | 13.8 | 16.1 | 16.1 | 0.20 | 51.0 | 76.1 | 45.9 |
| | UPop | 14.0 | 16.1 | 16.5 | 0.16 | 50.7 | 76.4 | 42.1 |
| | EMLoC | 14.2 | 16.1 | 17.8 | 0.16 | 52.3 | 85.2 | 48.8 |
| 25% | InternVL 2B | 10.9 | 4.9 | 12.9 | 0.29 | 44.6 | 78.1 | 34.4 |
| | Offsite | 10.7 | 16.1 | 9.3 | 0.33 | 50.6 | 76.6 | 45.4 |
| | UPop | 11.3 | 16.1 | 10.3 | 0.22 | 50.9 | 76.6 | 44.1 |
| | EMLoC | 11.5 | 16.1 | 11.3 | 0.21 | 51.6 | 79.6 | 46.2 |

Table 12: **Overall time overhead analysis.** We compare the time cost of EMLoC in each stage with
LoRA and QLoRA. Note that due to our design choice of training free emulator construction and
LoRA correction stage, the time overhead introduced by these additional stage is negligible.

| Size | Method | Construction (hr) | Fine-tuning (hr) | Correction | #GPU | Overall wall time (hr) |
|------|--------|-------------------|------------------|------------|------|------------------------|
| 8B | LoRA | 0 | 11.6 | 0 | 1 | 11.6 |
| | QLoRA | 0 | 12.1 | 0 | 1 | 12.1 |
| | EMLoC | 0.3 | 4.7 | 20 (sec.) | 1 | 5.0 |
| 38B | LoRA | 0 | 15.8 | 0 | 4 | 15.8 (estimated) |
| | QLoRA | 0 | 19.5 | 0 | 2 | 19.5 |
| | EMLoC | 1.2 | 14.2 | 50 (sec.) | 1 | 15.6 |

While surpassing zero-shot performance is not surprising, it is important to highlight that EMLoC
achieves these gains under the same memory requirements as inference. This means that users who
are able to run inference on these models can also benefit from task-specific fine-tuning—without
needing additional memory resources—making model adaptation more accessible in practice.

## 11.2 Computation Resource

We report the computational resource usage of our main experiments in Table 11, including memory
consumption during both fine-tuning and inference. Additionally, we present the FLOPs per forward
pass and the throughput, measured as the time taken for a full forward and backward pass on a single
training sample. All metrics are evaluated on the same machine with a sequence length of 2048
to ensure fair comparison. Notably, Offsite-tuning exhibits slightly higher throughput due to its
emulator using significantly fewer layers than other baselines. Despite this, EMLoC achieves superior
fine-tuning performance while maintaining comparable computational resource, demonstrating its
efficiency and effectiveness.

For a more comprehensive analysis, we include additional wall-clock runtime comparisons in Table 12,
covering the full EMLoC pipeline — including emulator construction, fine-tuning, and LoRA
correction — against both LoRA and QLoRA fine-tuning. From these results, we observe that EMLoC
achieves lower overall runtime compared to the baselines, even when accounting for the overhead
of emulator construction and correction. This efficiency arises from the activation-aware SVD
construction, which is fast and avoids expensive continual pretraining, as well as from the lightweight
emulator, which improves throughput during fine-tuning. Moreover, the correction step incurs
negligible computational cost relative to other stages while providing additional performance gains.
These findings reinforce the practical impact of EMLoC: it not only reduces memory consumption
but also remains competitive or superior in wall-clock efficiency, making it well-suited for real-world,
resource-constrained deployment scenarios.

Table 13: **Comparison between EMLoC and QLoRA.** Quantization-based methods offer limited flexibility in memory reduction and can degrade inference performance, while EMLoC enables fine-tuning on consumer-grade 24GB GPUs without affecting the base model's inference quality.

| Method | InternVL2.5-26B | | InternVL2.5-38B | |
|--------|-----------------|--|-----------------|--|
| | Fine-tuning memory (GB) | WC VQA | Fine-tuning memory (GB) | WC VQA |
| Zero-shot | - | 53.6 | - | 51 |
| LoRA | 72.9 | - | 94.8 | - |
| QLoRA | 38.1 | 39.0 | 43.4 | 43.6 |
| EMLoC | **18.6** | **56.8** | **20.1** | **52.6** |

Table 14: **Compare EMLoC with stronger small model**.

| Model | Method | Fine-tuning memory | PMC-VQA | WebSRC (val set) | WC-VQA |
|-------|--------|--------------------|---------|------------------|--------|
| InternVL 2.5 2B | LoRA | 10.9 (GB) | 44.6 | 78.2 | 34.4 |
| InternVL 3 2B | LoRA | 10.9 (GB) | 48.4 | 79.5 | 44.8 |
| InternVL 2.5 8B | EMLoC | 11.5 (GB) | 51.6 | 80.8 | 46.2 |

## 11.3 Comparison with QLoRA

Although quantization-based methods like QLoRA are orthogonal to our focus—they do not reduce fine-tuning memory relative to inference, we include a comparison to highlight the potential advantages of our emulator-based approach. Results are presented in Table 13. Memory usage is measured in the same way as Table 11, with the exception that methods other than EMLoC are run across multiple GPUs without distributed data parallelism (DDP), and model weights are split across devices. Due to limited computing resources, we omit standard LoRA fine-tuning results for these large models.

By design, quantization methods offer limited flexibility and a fixed upper bound on memory savings. As shown in Table 13, even with 4-bit quantization, QLoRA cannot support fine-tuning of InternVL2.5-26B or InternVL2.5-38B on a 24GB consumer GPU. In contrast, EMLoC enables flexible memory reduction via emulator construction, allowing fine-tuning under memory budgets. Furthermore, while quantization performs well in most cases, it can degrade the original model's performance—even during inference. In our experiments, quantizing InternVL2.5-26B and 38B significantly reduced accuracy, to the extent that QLoRA underperformed compared to the half-precision zero-shot baseline. These results underscore the practical benefits of emulator-based fine-tuning, offering greater memory flexibility without compromising inference performance.

## 11.4 Using Stronger Small Model

To provide further insight, we conducted an additional experiment using InternVL3-2B (the latest version of the InternVL2.5 models used in main paper) as a stand-in for a compact model baseline. The results in Table 14 show that EMLoC applied to InternVL2.5-8B with a 25% compression ratio still outperforms direct fine-tuning of InternVL3-2B. This suggests that EMLoC can retain the performance benefits of larger models even under tight memory constraints, offering a compelling alternative to using smaller models.

## 11.5 Applying EMLoC to Houlsby Adapter

Note that EMLoC's first two stages are fully compatible with other adapters, still enabling memory-efficient training. To demonstrate this, we conducted an additional experiment using Houlsby Adapters [13] in place of LoRA. As shown in Table 15, EMLoC still performs better baseline methods in Table 1, highlighting its flexibility beyond LoRA. Furthermore, we observe that the LoRA-based variant benefits noticeably from the correction algorithm, achieving higher performance than the Houlsby counterpart. This demonstrates the importance of the correction stage in enhancing LoRA's effectiveness within the EMLoC framework. However, the current LoRA correction algorithm is indeed tailored to the linearity of LoRA modules, and we believe there is no straightforward extension to other PEFT methods.

Table 15: **Combining EMLoC with Houlsby Adapter**. Note that the first 2 stage of EMLoC is compatible with other PEFT methods to achieve significant memory reduction.

| Method | PMC-VQA | WebSRC (val set) | WC-VQA |
|---|---|---|---|
| EMLoC w/o correction | 51.5 | 79.2 | 45.8 |
| EMLoC w/ correction | 51.6 | 80.8 | 46.2 |
| Houlsby Adapter | 51.2 | 79.0 | 44.5 |

## 11.6 Ablation Study on the Number of Calibration Data

EMLoC relies on a small calibration set $\mathcal{D}_C$ to perform activation-aware SVD during emulator construction. In our main experiments, we use 64 samples as the default size for $\mathcal{D}_C$. To evaluate the sensitivity to this choice, we vary the number of calibration examples and report the results in Fig. 5. As shown, model performance plateaus after using 32–64 calibration samples, indicating that only a small amount of task-specific data is sufficient to construct an effective emulator. This highlights the practicality of EMLoC in low-resource scenarios.

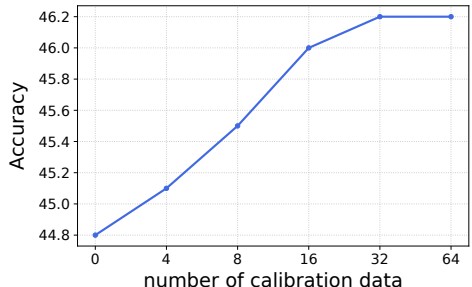

Figure 5: We plot performance on WC-VQA under different number of calibration data.

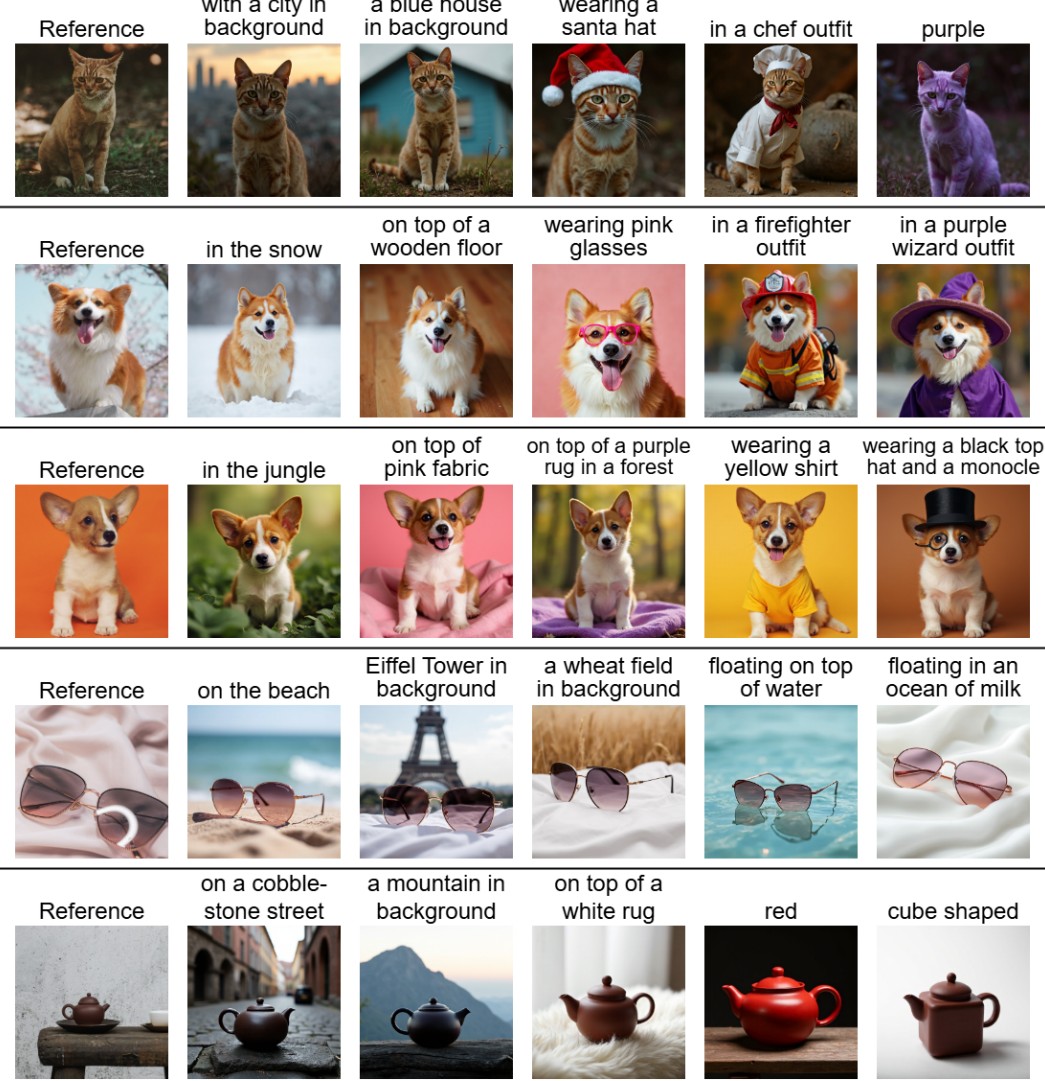

Figure 6: **Qualitative results of applying EMLoC to diffusion model personalization.** DreamBooth with LoRA is used to personalize the 12B FLUX.1-dev diffusion model, illustrating that EMLoC can be effectively extended to generative tasks beyond text.

