# OpenReview forum: "EMLoC: Emulator-based Memory-efficient Fine-tuning with LoRA Correction"
_NeurIPS.cc/2025/Conference — NeurIPS 2025 poster_

### Official Review · Reviewer_XKMB · 2025-06-21

**Clarity:** 2
**Significance:** 2
**Originality:** 2
**Rating:** 4
**Confidence:** 4

**Summary:**

This paper presents EMLoC, a three-stage framework that enables fine-tuning of large pre-trained models under the same GPU memory budget required for inference. First, it constructs a downstream-aware emulator by applying activation-aware SVD to each weight matrix using a small calibration set, yielding a lightweight low-rank model tailored to the target task. Second, it fine-tunes this emulator via LoRA, dramatically reducing memory overhead during training. Third, it introduces a novel LoRA correction algorithm that compensates for the misalignment between the emulator and the original model by adjusting the LoRA factors before merging them back into the full model. Experiments on multimodal VQA benchmarks (ChartQA, DocVQA, InfoVQA, TextVQA, PMC-VQA, WebSRC, WC-VQA) and NLP tasks (MathQA, GSM8K) demonstrate that EMLoC consistently outperforms strong baselines—including Offsite-Tuning, row-pruning (UPop), and LORAM—under 25–50% memory budgets, and scales to 26B and 38B models on a single 24 GB GPU.

**Questions:**

Formatting: In table 5, the last row reads "Ours" while others reads "EMLoC." Perhaps this is a typo.

More Baselines: It would be great to see more baselines on distillation or smaller foundation models when comparing EMLoC.

Time Costs: Apart from memory cost, it is important to also evaluate the time cost of EMLoC. It would be less practical if the efficiency in memory usage comes at significant cost on runtime.

**Ethical Concerns:**

["NO or VERY MINOR ethics concerns only"]

**Final Justification:**

The authors have highlighted the runtime cost and demonstrated the improvements from EMLoC during rebuttal, which I found to be convincing.

**Limitations:**

yes

**Paper Formatting Concerns:**

no concerns

**Quality:**

2

**Strengths And Weaknesses:**

Strengths
Memory-Efficiency Guarantee: By construction, EMLoC closes the memory gap between inference and fine-tuning without requiring specialized hardware or gradient checkpointing techniques. Users can adapt models as large as 38 B with only 24 GB of GPU RAM.

Downstream-Aware Compression: Activation-aware SVD ensures the emulator retains task-relevant features, outperforming both generic SVD and row-pruning approaches while incurring minimal construction overhead (~0.3 GPU-hours vs. >200 GPU-hours).

Correction Mechanism: The proposed correction algorithm rigorously derives a basis (the SVD factors of the LoRA input matrix), computes the emulator→model output discrepancy, and clamps corrections to preserve stability—yielding substantial accuracy gains (e.g., +3–4 points over finetuned emulator alone).

Comprehensive Evaluation: The paper evaluates across seven VQA datasets (including challenging domain-specific ones) and two math reasoning benchmarks, as well as multiple model scales (8 B→38 B), demonstrating consistent gains and strong robustness to hyperparameters.

Weaknesses:

Compute Overhead Details: Although emulator construction is fast, the paper lacks explicit wall-clock or FLOP comparisons for the fine-tuning and correction stages versus baselines, which would clarify real-world training costs.

Time Efficiency: EMLoC ensures efficient usage of memory at training and inference, but there lacks analysis on the time costs of EMLoC.

Other Distillation Baselines: A more appropriate baseline would be a foundation model of smaller size that is capable to be finetuned within the memory budget. There lacks comparison in this aspect, and the only direct finetuning baseline is from a larger (infeasible) model and serves as an upperbound.

---

> ### Author Rebuttal · Authors · 2025-07-31
>
> **The paper lacks explicit analysis of time costs (e.g., wall-clock or FLOP) of EMLoC for the fine-tuning and correction stages versus baselines, which would clarify real-world training costs. (W1, W2, Q3)**
>
> We thank the reviewers for raising such practical concerns. We have provided detailed measurements of memory consumption, throughput, and FLOPs during fine-tuning in Table 10 of the Supplementary Material, and the overall overhead of emulator construction in Table 4. For a more comprehensive analysis, we now include additional wall-clock runtime comparisons covering the full EMLoC pipeline — including emulator construction, fine-tuning, and LoRA correction — against both LoRA and QLoRA fine-tuning.
>
> |Size|Method|Construction (hr)|Fine-tuning (hr)|Correction|#GPU|Overall wall time (hr)|
> |:-:|:-:|:-:|:-:|:-:|:-:|:-:|
> |8B|LoRA|0|11.6|0|1|11.6|
> |8B|QLoRA|0|12.1|0|1|12.1|
> |8B|EMLoC|0.3|4.7|20 sec|1|5.0|
> |38B|LoRA|0|15.8|0|4|15.8 (estimated)|
> |38B|QLoRA|0|19.5|0|2|19.5|
> |38B|EMLoC|1.2|14.2|50 sec|1|15.6|
>
> From these results, we observe that:
>
> - EMLoC achieves lower overall runtime compared to the baselines, even when accounting for the overhead of emulator construction and correction.
> - This efficiency stems from the activation-aware SVD construction, which is fast and avoids expensive continual pretraining, and from the lightweight emulator, which increases throughput during fine-tuning.
> - Furthermore, our correction step is computationally negligible compared to other stages while providing additional performance improvement.
>
> These findings reinforce the practical impact of EMLoC: it not only reduces memory consumption but also remains competitive or superior in wall-clock efficiency, making it well-suited for real-world, resource-constrained deployment scenarios.
>
> ----
> **Lack of comparisons with baselines on distillation or a foundation model of smaller size that is capable of being finetuned within the memory budget. (W3, Q2)**
>
> Thanks for the insightful suggestion. In fact, we have already included such comparisons in Table 1, where we report results for InternVL2B and InternVL4B, both of which have comparable training costs with our methods.
>
> Regarding the suggestion to include distilled model baselines, we appreciate this valuable point. Unfortunately, we were unable to find any open-source, well-distilled versions of the InternVL2.5 series for a fair and direct comparison. Additionally, due to resource constraints, we are not in a position to conduct large-scale distillation ourselves. Nevertheless, to provide further insight, we conducted an additional experiment using InternVL3-2B (the latest version of the InternVL2.5 models used in our paper) as a stand-in for a compact model baseline. The results below show that EMLoC applied to InternVL2.5-8B with a 25% compression ratio still outperforms direct fine-tuning of InternVL3-2B. This suggests that EMLoC can retain the performance benefits of larger models even under tight memory constraints, offering a compelling alternative to using smaller or distilled models.
>
> | Model           | Method | Fine-tuning memory (GB) | PMC-VQA | WebSRC (val set) | WC-VQA |
> |:-----------------:|:--------:|:-------------------------:|:---------:|:------------------:|:--------:|
> | InternVL 2.5 2B | LoRA   | 10.9                    | 44.6    | 78.2             | 34.4   |
> | InternVL 3 2B   | LoRA   | 10.9                    | 48.4    | 79.5             | 44.8   |
> | InternVL 2.5 8B | EMLoC  | 11.5                    | 51.6    | 80.8             | 46.2   |
>
> ---
> **Formatting: In table 5, the last row reads "Ours" while others read "EMLoC." Perhaps this is a typo. (Q1)**
>
> We thank the reviewer for catching this. Yes, the last row in Table 5 should read "EMLoC", and we will correct it in the final version.

---

> > ### Comment · Reviewer_XKMB · 2025-07-31
> >
> > Thank you for your response. They have addressed my concerns. I will increase my score to weak accept.

---

> > > ### Author Response · Authors · 2025-08-01
> > >
> > > Thank you for your valuable review and for taking the time to read our responses.
> > >
> > > We appreciate that our rebuttal addressed your concerns and will incorporate the clarifications and suggested references into our revised version.
> > >
> > > Thank you again for your constructive feedback.

---

### Official Review · Reviewer_VfBn · 2025-06-30

**Clarity:** 2
**Significance:** 3
**Originality:** 2
**Rating:** 4
**Confidence:** 3

**Summary:**

EMLoC introduces a three-stage pipeline for fine-tuning large models under the same memory budget as inference. First, it constructs a lightweight task-specific emulator by applying activation-aware SVD to the original model's weight matrices using a small calibration dataset. This yields a low-rank model that retains the same layer shapes but has far fewer parameters. Second, EMLoC fine-tunes only this emulator with standard LoRA adapters using any training pipeline. Finally, to deploy the full model, a novel LoRA correction algorithm adjusts the learned adapters to compensate for differences between the emulator and the original model. Inference then runs on the full model with the corrected LoRA modules. Experiments across vision–language and NLP tasks show that EMLoC matches or exceeds baselines despite having much tighter memory, and EMLoC achieves accuracy close to that of full-model tuning.

**Questions:**

1.	The method uses only 64 calibration examples for SVD. How sensitive are the results to this choice? Have the authors tested larger or differently distributed calibration sets? (Fig.4 shows slight $\lambda$ sensitivity, but not data-scale sensitivity.)

2.	Did the authors compare against the recent LoRAM method or evaluate quantization-based fine-tuning? How would EMLoC perform relative to these approaches, and what are the trade-offs?

3.	The correction algorithm is defined for linear LoRA modules $\Lambda = W_A W_B$. Can this idea extend to other adapter forms (e.g. prefix-tuning or Mixture-of-Adapters)? Which layers receive correction, and is $\lambda$ chosen per-layer or globally?

4.	What is the runtime and memory overhead of building the emulator (many SVDs) and performing LoRA correction? For very large models (38B+), is this overhead negligible compared to full training, or does it pose practical limits?

5.	The paper notes EMLoC works without quantization. Could the authors combine EMLoC with model quantization (e.g. 4-bit LoRA) to push memory even lower? Are there reasons this was not explored or any pitfalls?

Other limitations:

Memory Factors Not Addressed: EMLoC focuses on model-weight memory only. Optimizer states and activations can still dominate overall usage. Thus EMLoC alone may still require techniques like sparse Adam or activation checkpointing for very deep networks. In contrast, methods like Quantized Side Tuning tackle these sources as well.

Assumption of Linear Corrections: The LoRA correction enforces consistency of linear merged outputs on a subspace. This assumes the mismatch between full and low-rank models can be fixed by additive linear terms. If the compression-induced error is highly nonlinear, the correction might be insufficient, but the paper offers limited analysis of this.

Evaluation Scope: Most experiments are on VQA and math QA benchmarks. It is unclear how EMLoC performs on large-scale open-domain NLP tasks (e.g. summarization, translation) or on very low-data regimes. Also, diffusion results show somewhat lower image-similarity scores than full training, suggesting possible quality trade-offs not deeply examined.


References:

[1] Wang, Xin, et al. "Svd-llm: Truncation-aware singular value decomposition for large language model compression." arXiv preprint arXiv:2403.07378 (2024).

[2] Zhang, Jun, et al. "Train small, infer large: Memory-efficient lora training for large language models." arXiv preprint arXiv:2502.13533 (2025).

[3] Dettmers, Tim, et al. "Qlora: Efficient finetuning of quantized llms." Advances in neural information processing systems 36 (2023): 10088-10115.

[4] Xiao, Guangxuan, Ji Lin, and Song Han. "Offsite-tuning: Transfer learning without full model." arXiv preprint arXiv:2302.04870 (2023).

[5] Zhang, Zhengxin, et al. "Quantized side tuning: Fast and memory-efficient tuning of quantized large language models." arXiv preprint arXiv:2401.07159 (2024).

[6] Shi, Dachuan, et al. "UPop: Unified and progressive pruning for compressing vision-language transformers." International Conference on Machine Learning. PMLR, 2023.

**Ethical Concerns:**

["NO or VERY MINOR ethics concerns only"]

**Final Justification:**

the author has clarified my doubts.

**Limitations:**

yes

**Quality:**

2

**Strengths And Weaknesses:**

Strengths:

EMLoC directly eliminates the fine-tuning vs inference memory gap by training on a much smaller model. Its emulator preserves task-relevant knowledge via activation-aware SVD, and supports arbitrary LoRA placements. The LoRA correction is a novel, lightweight fix for model mismatch, enabling near upper-bound performance on held-out tasks. The paper is also well-presented with precise figures and ablations.

Weaknesses:

The core idea of fine-tuning on a compressed model and then transferring to the full model is similar to recent work. EMLoC's novelty lies in using SVD-LLM for the emulator and a specific correction, but it builds on prior concepts of adapter transfer. The emulator relies on a small calibration set, and the efficacy of the SVD approximation without whitening (unlike SVD-LLM) may limit fidelity. Only 64 calibration examples are used. It's unclear how sensitive the results are to this choice.

Meanwhile, LoRA correction involves hyperparameters ($\lambda$) and a basis choice, but its effect is only briefly analyzed (Fig. 4 suggests that the results are stable). The method only targets model-weight memory, indicating that it does not reduce optimizer state or activation memory; therefore, it remains complementary to approaches like gradient checkpointing or 4-bit quantization. Baselines omit some solid or recent alternatives (e.g., there is no direct comparison to the LoRAM scheme or quantized fine-tuning, such as QLoRA).
Finally, the algorithmic complexity of performing many SVDs and corrections on large models is not discussed; this might pose a practical overhead.

---

> ### Author Rebuttal · Authors · 2025-07-31
>
> **Novelty lies in using SVD-LLM for the emulator and a specific correction, but builds on prior concepts of adapter transfer. (W1)**
>
> We’re happy to clarify the novelty of our EMLoC and explain how it differs from prior SVD-based methods. Our EMLoC introduces several distinct innovations:
> 1. Flexibility in parameters to update: Unlike prior methods such as LoRAM (row pruning) and Offsite-Tuning (layer dropping), which impose hard constraints on which parameters can be updated, EMLoC adopts SVD to preserve the flexibility of standard fine-tuning, which allows all weights to be tunable.
> 2. No additional data / continual pretraining for emulator construction: By leveraging downstream data for activation-aware SVD, EMLoC avoids costly continual pretraining required by earlier approaches such as LoRAM, significantly reducing the overhead of emulator construction.
> 3. LoRA correction mechanism: We propose a LoRA correction mechanism that explicitly addresses the misalignment between the emulator used during training and the original model used during inference—a challenge that prior work has overlooked.
>
> As demonstrated in Table 4, these innovations enable EMLoC to achieve better performance (up to 2.6% in accuracy) with significantly lower overhead (0.3 vs. 214 GPU-hours) compared to prior methods like LoRAM.
>
> ---
> **How sensitive are the results to the size or the distribution of the calibration data? (W2, Q1)**
>
> As shown in Figure 5 of the Supplementary Materials, performance plateaus after 32–64 calibration samples. We therefore use 64 examples as a practical default, indicating that our method is not highly sensitive to calibration set size, and a very small number of downstream samples is sufficient for effective emulator construction.
>
> Regarding data distribution, Table 4 highlights the importance of using downstream data for calibration. Specifically, following LoRAM, we applied compression and continual pretraining using a general-domain dataset. However, this setup yielded worse results—up to 5% lower accuracy—compared to EMLoC where a small calibration set from the downstream-task dataset is used. This supports our design choice of using a small but downstream-aware sample set.
>
> ---
> **LoRA correction involves hyperparameters (𝜆) and a basis choice, but its effect is only briefly analyzed. (W3)**
>
> The impact of the 𝜆 is analyzed in Figure 4. We observe that increasing 𝜆 up to a moderate value (e.g., 5 in Figure 4) improves accuracy by 0.5%, likely due to less distortion in the correction term. However, overly large 𝜆 values degrade performance—even falling below that of uncorrected LoRA. We hypothesize that an overly large correction term may dominate the original LoRA, ultimately degrading performance.
>
> Regarding the basis choice in the correction step, we provide a detailed discussion in Section C.2 of the Supplementary Material. We show that the resulting corrected LoRA weights remain the same, regardless of the choice of basis—demonstrating that the correction is mathematically invariant to this choice.
>
>
> ---
> **EMLoC focuses on model-weight memory only, not reducing optimizer state or activation memory. It remains complementary to approaches like gradient checkpointing or 4-bit quantization. (W4, L1)**
>
> It is correct that EMLoC focuses on reducing memory related to model weights. However, our method can be served as complementary to existing techniques. In fact, EMLoC is designed to be integrated seamlessly with methods like quantization, and activation checkpointing. As shown in Table 1, we combine EMLoC with QLoRA, and gradient checkpointing, and consistently observe better performance than baselines such as row pruning proposed by LoRAM and layer dropping proposed by Offsite-Tuning. These results demonstrate that EMLoC effectively reduces the memory footprint of model weights while remaining fully compatible with other techniques, jointly advancing the frontier of memory-efficient fine-tuning.
>
> ---
> **Did the authors compare against the recent LoRAM method or evaluate quantization-based fine-tuning? (W5, Q2)**
>
> We have included comparisons and discussion with QLoRA, using InternVL2.5 26B and 38B models in Table 11 of the Supplementary Material, where we observe that even with 4-bit quantization, QLoRA still requires 43GB of memory—while EMLoC can fine-tune the 38B model using only 20GB and achieves up to a 9% accuracy gain. Compared to QLoRA, EMLoC offers two main advantages:
> - Greater flexibility in controlling the compression ratio, rather than being constrained by fixed bit-widths in quantization-based method;
> - No degradation in inference performance, since quantization affects the model even during inference, whereas EMLoC retains the original model parameters for deployment
>
> Regarding LoRAM, we clarify that our experiments do include a direct comparison. As shown in Table 4, the first row corresponds to LoRAM, while the last row corresponds to EMLoC. We observe that EMLoC achieves comparable or better performance, while also incurring significantly lower overhead for constructing the emulator.
>
> ---
> **Could the authors combine EMLoC with model quantization to push memory even lower? (Q5)**
>
> Yes, we have explored combining EMLoC with quantization, and the results are presented in Table 1, specifically in the third block of rows. The results show that EMLoC still outperforms prior baselines—row pruning corresponding to LoRAM, and layer dropping corresponding to Offsite-Tuning—by up to 2.9% in accuracy, suggesting that EMLoC is complementary to quantization for further memory reduction.
>
> ---
> **What is the runtime and memory overhead of building the emulator and performing LoRA correction? (W6, Q4)**
>
> We thank the reviewers for raising such practical concerns. We have provided detailed measurements of memory consumption, throughput, and FLOPs during fine-tuning in Table 10 of the Supplementary Material, and the overall overhead of emulator construction in Table 4. For a more comprehensive analysis, we now include additional wall-clock runtime comparisons covering the full EMLoC pipeline — including emulator construction, fine-tuning, and LoRA correction — against both LoRA and QLoRA fine-tuning.
>
> |Size|Method|Construction (hr)|Fine-tuning (hr)|Correction|#GPU|Overall wall time (hr)|
> |:-:|:-:|:-:|:-:|:-:|:-:|:-:|
> |8B|LoRA|0|11.6|0|1|11.6|
> |8B|QLoRA|0|12.1|0|1|12.1|
> |8B|EMLoC|0.3|4.7|20 sec|1|5.0|
> |38B|LoRA|0|15.8|0|4|15.8 (estimated)|
> |38B|QLoRA|0|19.5|0|2|19.5|
> |38B|EMLoC|1.2|14.2|50 sec|1|15.6|
>
> The table below details EMLoC's memory usage across different steps. For the 8B model, each step of EMLoC uses less than 12GB, while the 38B model stays under 24GB. This efficiency makes EMLoC accessible to individual researchers, as it avoids the substantial computational and memory overhead typically required by previous methods that rely on continual pretraining.
> |Model size|Construction (GB)|Fine-tuning (GB)|Correction (GB)|
> |:-:|:-:|:-:|:-:|
> |8B|10.8|11.5|0.7|
> |38B|23.8|20.1|0.8|
>
> ---
> **Most experiments are on VQA and math QA benchmarks. How does  EMLoC perform on large-scale open-domain NLP tasks or on very low-data regimes? (L3)**
>
> Thanks for the suggestion. While recent works such as LoRAM (ICLR 2025) have primarily focused on NLP tasks like QA and code generation, our evaluation goes beyond standard text-generation settings by including vision-language tasks (VQA) and diffusion model customization, demonstrating the broad applicability of EMLoC. Due to time constraints during the rebuttal period, we were unable to include additional experiments in other domains, but we have conducted more extensive ablations and added new baselines to address the review comments and to strengthen our analysis.
>
> ---
> **Can EMLoC extend to other adapter forms, considering that the correction algorithm is defined for LoRA? (Q3)**
>
> Yes. EMLoC’s first two stages are fully compatible with other adapters, still enabling memory-efficient training. To demonstrate this, we conducted an additional experiment using Houlsby Adapters in place of LoRA. As shown in the table below, EMLoC still performs better baseline methods in Table 1, highlighting its flexibility beyond LoRA. However, the current LoRA correction algorithm is indeed tailored to the linearity of LoRA modules, and we believe there is no straightforward extension to other PEFT methods.
>
> |Method|PMC-VQA|WebSRC (val set)|WC-VQA|
> |:-:|:-:|:-:|:-:|
> |LoRA w/o correction|51.5|79.2|45.8|
> |LoRA w/ correction|51.6|80.8|46.2|
> |Houlsby Adapter|51.2|79.0|44.5|
>
> ---
> **Which layers receive correction, and is chosen per-layer or globally? (Q3)**
>
> The correction algorithm is applied to each LoRA independently, and we will clarify this in the revised paper.
>
> ---
> **The LoRA correction assumes linear mismatch between the emulator and the original model on a subspace. What if the mismatch is nonlinear? (L2)**
>
> We agree with the reviewer that our correction method assumes a linear correction. However, this limitation is shared by all approaches that apply weight-based corrections to LoRA modules without modifying the model architecture. As noted in (W1), EMLoC advances beyond prior work by explicitly addressing the mismatch between the training and inference models—an issue previously overlooked. While nonlinear correction remains an open direction for all LoRA-based methods, our results show that even linear correction consistently improves performance.
>
> ---
> **Diffusion results show somewhat lower scores than full training, suggesting possible quality trade-offs not deeply examined. (L4)**
>
> We totally agree with this observation. All efficient fine-tuning methods, including ours, involve trade-offs between efficiency and performance. While EMLoC achieves significant memory savings, there may be slight drops in performance compared to our upper-bound, full training.

---

> > ### Comment · Reviewer_VfBn · 2025-08-06
> >
> > I have the follow-up questions. i) Apart from QLora, does the author try to combine EMLoC with other methods? 2) The author has stated the understandable tradeoffs between performance and memory savings. How does the tradeoff rank among other methods? 3) Is there an ablation showing the benefit of correction algorithm when applied to each LoRA independently vs globally?

---

> > > ### Author Response · Authors · 2025-08-07
> > >
> > > **Apart from QLora, does the author try to combine EMLoC with other methods?**
> > >
> > > Yes! As noted in our responses to (W4, L1), EMLoC is compatible with other memory-efficient techniques. Note that training memory can be decomposed into three main components: (1) *Optimizer state memory* (information such as momentum and variance in the Adam optimizer), (2) *Activation memory* (values retained during the forward pass for use in backpropagation), and (3) *Model weights memory* (the full base model for forward and backward passes). Specifically,  EMLoC targets at (3) *model weights memory*. In Table 1 of our paper, we've combined EMLoC with
> > >
> > > - Gradient checkpointing [3]: Reduces (2) *activation memory* by saving fewer intermediate values during the forward pass.
> > > - QLoRA [8]: Reduces (1) *optimizer state* via LoRA and compresses model weights via quantization.
> > >
> > > As can be seen in Table 1, we compared the ability of EMLoC to combine with baseline methods corresponding to LoRAM [41], which also aim to reduce (3) *model weight memory*. In a setting that reduces both *model weight* and *activation memory*, EMLoC + gradient checkpointing outperformed LoRAM + gradient checkpointing by 3% (79.6% vs. 76.6%). In a more aggressive setting that reduces all three components of training memory, EMLoC + gradient checkpointing + QLoRA outperformed LoRAM + gradient checkpointing + QLoRA by 2.3% (78.1% vs. 75.8%). These results highlight EMLoC’s compatibility and effectiveness in conjunction with diverse memory-saving strategies.
> > >
> > > ---
> > > **The author has stated the understandable tradeoffs between performance and memory savings. How does the tradeoff rank among other methods?**
> > >
> > > Thanks for raising this issue. In fact, we have provided direct comparisons with methods targeting *model weight memory* reduction such as LoRAM [41] and QLoRA [8] in our paper. In Tables 3, under the same memory usage of 12.6GB, EMLoC outperformed LoRAM [41] by 2.6% (29.8% vs. 27.2%) on Math QA tasks, and in Table 4, achieved a 2.6% gain (46.2% vs. 43.6%) on VQA tasks. The comparison with QLoRA is shown in Table 11, where we observe EMLoC outperformed QLoRA by 9% (43.6% vs. 53.6%) on VQA tasks while using significantly lower memory usage (20.1GB vs. 43.4GB). These results demonstrate that EMLoC offers a more favorable trade-off between performance and memory savings on model weights compared to existing methods.
> > >
> > > Comparisons with methods targeting other components of training memory (i.e., (1) *optimizer state memory* or (2) *activation memory*) are more difficult, as one cannot easily set equivalent memory savings across different techniques. However, as mentioned in our responses to (W4, L1) and the previous follow-up question, EMLoC can be directly integrated with methods targeting other components (e.g., EMLoC + gradient checkpointing or EMLoC + gradient checkpointing + QLoRA), enabling joint use to achieve further memory savings while preserving accuracy.
> > >
> > > ---
> > > **Is there an ablation showing the benefit of the correction algorithm when applied to each LoRA independently vs globally?**
> > >
> > > We did not develop a global correction algorithm to compensate for LoRAs across multiple layers, since our correction relies on the linearity of individual LoRA modules (see our prior response to (L2)). Applying correction globally would require modeling across intervening non-linear layers, which breaks this assumption and makes a global correction intractable. That said, as noted in our response to (L2), (W6) and (W5) of Reviewer ZRd4, our independent and linear correction is a training-free technique, which consistently improves performance with minimal time overhead (under 1 minute), demonstrating its practical effectiveness within current design constraints.

---

> > > > ### Comment · Reviewer_VfBn · 2025-08-07
> > > >
> > > > I thank the authors for their clarification and will raise my score to WA.
> > > > -cheers.

---

> > > > > ### Author Response · Authors · 2025-08-07
> > > > >
> > > > > Thank you for your valuable review and for taking the time to read our responses.
> > > > >
> > > > > We're glad that our responses were able to address your concerns and we greatly appreciate your suggestions. We will be sure to incorporate the clarifications and suggested references into our revised version.
> > > > >
> > > > > Thank you again for your constructive feedback.

---

### Official Review · Reviewer_ZRd4 · 2025-06-30

**Clarity:** 3
**Significance:** 3
**Originality:** 3
**Rating:** 5
**Confidence:** 3

**Summary:**

This paper introduces a novel memory-efficient parameter-efficient fine-tuning method, with experimental results suggesting that it can effectively reduce GPU memory requirements during model fine-tuning. The authors claim that, without quantization, their method enables fine-tuning a 38B model on a consumer-grade 24GB GPU.

**Questions:**

See above

**Ethical Concerns:**

["NO or VERY MINOR ethics concerns only"]

**Final Justification:**

The author partially resolved my issue, and I will increase my rating.

**Limitations:**

See above

**Quality:**

3

**Strengths And Weaknesses:**

Strengths:​​

(1) The figures are well-organized, and Figure 3 appears particularly interesting.

(2) The experimental results are impressive.

(3) The motivation illustrated in Figure 1 is easy to follow.

​​Weaknesses:​​

(1) The writing in Section 3.1 is confusing. The section title defines a memory-efficient paradigm, yet the content seems to be a formalized description of the LoRA training framework. I did not find any discussion on memory efficiency in this section.

(2) I do not fully understand the role of the "emulator" in this paper or how it reduces training memory usage. How does it differ from LoRA? Could the authors provide a more detailed explanation?

(3) In Section 3.2, the authors list three necessary conditions for memory efficiency: (a) fewer parameters, (b) flexibility and tunability, and (c) retention of downstream task knowledge. These are inherently characteristics of LoRA itself. I am particularly curious about how these three points contribute to memory efficiency, especially regarding activation/optimizer states as depicted in Figure 1.

(4) The SVD-based design has already been explored in LoRand. The authors should discuss the differences between their method and LoRand in the related work section.

(5) The authors provide extensive textual descriptions of LoRA modifications, and Figure 3 looks intriguing. However, the results in Table 6 suggest that the impact of LoRA modifications is marginal. Do the authors have further explanations for this observation?

(6) The results in Table 4 are remarkable. However, after reading the paper, I still do not grasp why the proposed method significantly reduces training costs. Could the authors explain this in simpler terms? Additionally, refining the writing could help reduce the readers' comprehension burden.

(7) E3VA [1] (from MSRA) is a representative work in memory-efficient fine-tuning. Have the authors attempted to compare training costs with E3VA?

[1] Parameter-efficient is not sufficient: Exploring parameter, memory, and time efficient adapter tuning for dense predictions, ACMMM'24.

---

> ### Author Rebuttal · Authors · 2025-07-31
>
> **What is the difference and relation between EMLoC and previous works like LoRA and E3VA? (W1, W7)**
>
> To clarify, as explained in Lines 32–40, the total memory cost of fine-tuning includes three main components:
> - Optimizer states — auxiliary information such as momentum and variance in the Adam optimizer for each trainable parameter;
> - Intermediate activations — values retained during the forward pass for use in backpropagation;
> - Model parameters — the full base model itself for forward and backward passes.
>
> LoRA reduces memory usage by lowering the number of trainable parameters, thereby decreasing the optimizer states. E3VA reduces activation memory by avoiding backpropagation through the backbone via architectural changes to module placement. In contrast, EMLoC addresses the often-overlooked bottleneck—the memory required to load and store the overall model parameters—by constructing a lightweight emulator to replace the original model during fine-tuning. Thus, EMLoC targets a fundamentally different source of memory consumption compared to LoRA and E3VA. However, as shown in Table 1, EMLoC combined with QLoRA and gradient checkpointing still consistently outperforms baselines such as row pruning proposed by LoRAM and layer dropping proposed by Offsite-Tuning. These results demonstrate EMLoC’s compatibility with other techniques and effectiveness in reducing model weight memory
>
> ---
> **What is the role of the "emulator" in EMLoC? How does the emulator-based framework reduce training memory usage? What is the difference between emulator-based framework and the standard LoRA framework? (W1, W2, W3)**
>
> To reduce the memory usage of model parameters, EMLoC introduces a lightweight surrogate model (i.e., the emulator), to replace the original model during standard LoRA fine-tuning, as described in Section 3.1. This emulator is constructed via activation-aware SVD using a small calibration set with the following three criteria (discussed in Section 3.2):
>
> 1. Fewer parameters: This enables memory-efficient fine-tuning since EMLoC no longer needs to load full original model parameters during fine-tuning
> 2. Flexible placement of LoRA modules: This makes EMLoC fully compatible with standard LoRA workflows and serves as a plug-and-play replacement.
> 3. Retention of downstream task knowledge: This makes the emulator behave closer to the original model during fine-tuning, alleviating the model gap between the emulator and the full model.
>
> This design differs from standard LoRA, which only reduces the number of trainable parameters but still requires loading the full model into memory. In contrast, EMLoC reduces the memory footprint of the entire model during training.
>
> ---
> **The results in Table 4 are remarkable, but it is not clear why the proposed method significantly reduces training costs. (W6)**
>
> The significant reduction in emulator construction overhead shown in Table 4 comes from our use of activation-aware SVD on a small set of downstream data. Compared to prior methods like LoRAM which relies on continual pretraining over large general corpora and causes substantial overhead and potential downstream-task misalignment, our EMLoC is a lightweight and efficient approach. As a result, our method not only avoids this costly stage, but also produces an emulator more tailored to the target task, leading to both lower construction cost (0.3 vs. 214 GPU-hours) and better downstream performance (up to 2.6% difference in accuracy).
>
> ---
> **Missing discussion on a SVD-based design of LoRand. (W4)**
>
> Unfortunately, we were unable to locate a method called “LoRand,” but we believe the reviewer may be referring to RandLoRA [1]. While RandLoRA also utilizes SVD, its goal is fundamentally different from ours. RandLoRA draws inspiration from SVD to reformulate the LoRA module as a summation of multiple bases in order to improve expressiveness. In contrast, our use of SVD is applied to the base model itself, with the objective of compressing model parameters to enable memory-efficient fine-tuning. We will clarify this distinction and cite RandLoRA in the related work section.
>
> [1] Albert, Paul, et al. "RandLoRA: Full-rank parameter-efficient fine-tuning of large models." arXiv preprint arXiv:2502.00987 (2025).
>
> ---
> **Table 6 seems to suggest that the impact of LoRA modifications is marginal. (W5)**
>
> We thank the reviewer for this observation. The marginal improvement is due to our aggressive compression setting, which may lead to relatively large misalignment between the emulator and the original model.
>
> To better understand the results in Table 6, we conducted an additional ablation experiment using a less aggressive 50% compression ratio during emulator construction. The results support our hypothesis that the effectiveness of LoRA correction is related to the degree of misalignment between the emulator and the original model. When the emulator is constructed without activation-aware SVD, it fails to preserve key downstream behavior of the original model, possibly resulting in greater misalignment during fine-tuning and, consequently, weaker correction effects. Conversely, with less severe compression, the misalignment is reduced, and we observe that the impact of LoRA correction becomes more pronounced.
>
> These findings suggest that while our current use of SVD-LLM provides a strong baseline for emulator construction, there is room for future exploration of more adaptive or task-aware compression strategies to further enhance alignment and correction effectiveness.
>
> | Ratio | Activation aware SVD | LoRA Correction | PMC-VQA | WebSRC | WC-VQA |
> |:-------:|:----------------------:|:-----------------:|:---------:|:--------:|:--------:|
> | 25%   | x                    | x               | 51.0    | 74.4   | 44.7   |
> | 25%   | x                    | v               | 51.2    | 74.4   | 44.8   |
> | 25%   | v                    | x               | 51.5    | 79.0   | 45.8   |
> | 25%   | v                    | v               | 51.6    | 79.6   | 46.2   |
> | 50%   | v                    | x               | 51.8    | 83.7   | 47.9   |
> | 50%   | v                    | v               | 52.3    | 84.9   | 48.8   |

---

> > ### Comment · Reviewer_ZRd4 · 2025-08-01
> > **About LoRand**
> >
> > Sorry for missing LoRand's paper title: 1% VS 100%: Parameter-Efficient Low Rank Adapter for Dense Predictions

---

> > > ### Author Response · Authors · 2025-08-02
> > >
> > > Thank you for providing the reference. LoRand can be viewed as a variant of the Houlsby adapter [2] that uses an SVD-like formulation to parameterize the up- and down-projection matrices. Specifically, it represents adapter weights as a product of summation of two low-rank matrices and a kernel matrix (i.e., $W=\sum P^TKQ$). By adjusting the rank and sharing kernel matrices across layers, LoRand further reduces the number of trainable parameters.
> > >
> > > Similar to the distinction between EMLoC and LoRA (as discussed in our response to W1), LoRand focuses on reducing the size of trainable adapter modules, whereas EMLoC reduces the memory for the pretrained weights in the base model, which improves the efficiency in a way not considered by LoRand. These approaches are orthogonal and complementary. We thank the reviewer again for suggesting a relevant work, and we will clarify this in the related work section and include a proper citation to LoRand.
> > >
> > > [2] Houlsby, Neil, et al. "Parameter-efficient transfer learning for NLP." International conference on machine learning. PMLR, 2019.

---

### Official Review · Reviewer_5hvp · 2025-07-02

**Clarity:** 4
**Significance:** 4
**Originality:** 4
**Rating:** 5
**Confidence:** 4

**Summary:**

This paper proposes EMLoC, a novel emulator-based fine-tuning framework. EMLoC constructs an emulator by applying low-rank compression to the model parameters. Fine-tuning is then performed on the emulator using LoRA, and the resulting LoRA weights are corrected before being applied back to the original model for inference. This approach significantly reduces both the training time and memory footprint.

**Questions:**

See above

**Ethical Concerns:**

["NO or VERY MINOR ethics concerns only"]

**Final Justification:**

The author resolved my issue, and I will keep my rating.

**Quality:**

4

**Strengths And Weaknesses:**

Strengths
+ The paper presents a clear motivation and a well-structured narrative. The proposed method is explained clearly and thoroughly.
+ The proposed method is practically meaningful and facilitates model fine-tuning under resource-constrained settings.
+ The method is novel, and the proposed correction technique offers valuable insights.

Weaknesses
+ Since QLoRA also reduces training costs, would it be possible to provide a direct comparison between EMLoC and QLoRA—both in terms of accuracy and memory consumption—to better demonstrate the advantages of EMLoC?
+ As a suggestion, I believe that EMLoC shares some conceptual similarities with module arithmetic, and it may be beneficial to include a discussion of this connection in the related work or appendix.

[1] Ziplora: Any subject in any style by effectively merging loras
[2] Token Compensator: Altering Inference Cost of Vision Transformer without Re-Tuning
[3] Lcm-lora: A universal stable-diffusion acceleration module

---

> ### Author Rebuttal · Authors · 2025-07-31
>
> **Provide a direct comparison between EMLoC and QLoRA in terms of accuracy and memory consumption, better demonstrating the advantages of EMLoC. (W1)**
>
> Thanks for the suggestion. In fact, we have included comparisons and discussion with QLoRA, using InternVL2.5 26B and 38B models in Table 11 of the Supplementary Material, where we observe that even with 4-bit quantization, QLoRA still requires 43GB of memory—while EMLoC can fine-tune the 38B model using only 20GB and achieves up to a 9% accuracy gain. Compared to QLoRA, EMLoC offers two main advantages:
> - Greater flexibility in controlling the compression ratio, rather than being constrained by fixed bit-widths in quantization-based method;
> - No degradation in inference performance, since quantization affects the model even during inference, whereas EMLoC retains the original model parameters for deployment
> ---
> **EMLoC shares some conceptual similarities with module arithmetic. It may be beneficial to include discussions of this connection in related work or appendix. (W2)**
>
> We thank the reviewer for pointing out the potential connection to module arithmetic. We agree that, at a high level, both approaches involve modifying or adjusting model weights. While there is a conceptual similarity, our LoRA correction method serves a different purpose. Module arithmetic techniques typically aim to merge multiple modules to combine different capabilities. In contrast, our method focuses on addressing the discrepancy between the compressed emulator used during training and the full model used during inference, by compensating for the misalignment introduced by this difference to further improve the performance. We will include the above discussions into the related work session.

---

> ### Comment · Reviewer_5hvp · 2025-08-05
>
> Thank you for your response. I will keep my score.

---

> > ### Author Response · Authors · 2025-08-05
> >
> > Thank you for your valuable review and for taking the time to read our responses.
> >
> > We appreciate that our rebuttal addressed your concerns and will incorporate the clarifications and suggested references into our revised version.
> >
> > Thank you again for your constructive feedback.

---

### Note · Authors · 2025-08-13

We thank the reviewers and AC for their constructive feedback. The rebuttal period has been exceptionally productive, allowing us to clarify critical aspects and validate the contributions of our work.

### Main Contribution and Strength
- **Same memory for fine-tuning as inference**: EMLoC enables fine-tuning with memory usage same as inference, unlocking adaptation without hardware beyond needed for deployment.
- **No additional data / pretraining for emulator construction**: By leveraging activation-aware SVD, EMLoC avoids continual pretraining required by previous works like LoRAM, significantly reducing the overhead (0.3 vs. 214 GPU-hours).
- **Novel LoRA correction mechanism**: We propose a training-free correction algorithm to recover misalignment between the emulator and original model, addressing a challenge overlooked by prior work and delivering consistent gains.

### Concerns Addressed during Rebuttal
- **Comparison with Other Methods**: EMLoC outperformed a SOTA of LoRAM by 2.6% (46.2% vs. 43.6%) with the same 11 GB memory (Table 4). In Table 11, EMLoC outperformed QLoRA by 9% (52.6% vs. 43.6%) while using significantly lower memory (20.1 vs. 43.4GB).\
[Addressing concerns from Reviewer 5hvp, VfBn]
- **Computation overhead**: Construction (0.3 hr) and correction (20 sec) incur negligible overhead, while the lightweight emulator enables higher throughput during fine-tuning, reducing overall wall-clock time to 5 hr—far less than LoRA (11.6 hr) and QLoRA (12 hr).\
[Addressing concerns from Reviewer VfBn, XKMB]
- **Additional experiments**: We added experiments comparing with new baselines (InternVL3), testing compatibility with the Houlsby adapter, and extending LoRA correction ablations. These results reinforce EMLoC’s effectiveness, flexibility with other PEFT modules, and consistent correction benefits.\
[Addressing concerns from Reviewer XKMB, ZRd4, VfBn]

### Final Remarks
Our results show that **EMLoC offers a better trade-off between memory savings and performance** compared to methods like quantization or LoRAM. Furthermore, it is **compatible with diverse memory-saving strategies** and practical to deploy.

Given the highlighted strengths, along with discussions provided during rebuttal, we believe this work will be of significant value to the efficient DL community. We will incorporate these points into the revised manuscript. We sincerely thank the AC and reviewers for their time and efforts in the review and decision-making process.

---

### Decision · Program_Chairs · 2025-09-17

**Decision:**

Accept (poster)

**Comment:**

This paper proposes a novel method called EMLoC for memory-efficient fine-tuning. The authors also provide a novel LoRA correction mechanism which accounts for misalignment between the original and distilled model. The proposed method enables fine-tuning a 38B model on 24GB GPU which is very efficient. All reviewers found experimental results convincing.

The authors provided a very detailed rebuttal addressing the majority of reviewers’ concerns. This includes providing new baseline comparison with QLoRA and verifying compatibility with other parameter-efficient fine-tuning methods. They also provided algorithmic complexity and sensitivity of calibration data size. I request that authors make sure all new experiments will be added in camera ready versions. Given initial positive feedback and detailed rebuttal I accept this paper.